# Anonymous Learning via Look-Alike Clustering: A Precise Analysis of Model Generalization

**Adel Javanmard**
University of Southern California, Google Research
ajavanma@usc.edu

**Vahab Mirrokni**
Google Research
mirrokni@google.com

## Abstract

While personalized recommendations systems have become increasingly popular, ensuring user data protection remains a top concern in the development of these learning systems. A common approach to enhancing privacy involves training models using anonymous data rather than individual data. In this paper, we explore a natural technique called *look-alike clustering*, which involves replacing sensitive features of individuals with the cluster's average values. We provide a precise analysis of how training models using anonymous cluster centers affects their generalization capabilities. We focus on an asymptotic regime where the size of the training set grows in proportion to the features dimension. Our analysis is based on the Convex Gaussian Minimax Theorem (CGMT) and allows us to theoretically understand the role of different model components on the generalization error. In addition, we demonstrate that in certain high-dimensional regimes, training over anonymous cluster centers acts as a regularization and improves generalization error of the trained models. Finally, we corroborate our asymptotic theory with finite-sample numerical experiments where we observe a perfect match when the sample size is only of order of a few hundreds.

## 1 Introduction

Look-alike modeling in machine learning encompasses a range of techniques that focus on identifying users who possess similar characteristics, behaviors, or preferences to a specific target individual. This approach primarily relies on the principle that individuals with shared attributes are likely to exhibit comparable interests and behaviors. By analyzing the behavior of these look-alike users, look-alike modeling enables accurate predictions for the target user. This technique has been widely used in various domains, including targeted marketing and personalized recommendations, where it plays a crucial role in enhancing user experiences and driving tailored outcomes [26, 19, 18, 21].

In this paper, we use look-alike clustering for a different purpose, namely to anonymize sensitive information of users. Consider a supervised regression setup where the training set contains $n$ pairs $(\boldsymbol{x}_i, \boldsymbol{y}_i)$, for $i \in [n]$, with $y_i \in \mathbb{R}$ denoting the response and $\boldsymbol{x}_i \in \mathbb{R}^d$ representing a high-dimensional vector of features. We consider two groups of features: sensitive features, which contain some personal information about users and should be protected from the leaner, and the non-sensitive features. We assume that that the learner has access to a clustering structure on users, which is non-private information (e.g. based on non-sensitive features or other non-sensitive data set on users).

We propose a look-alike clustering approach, where we anonymize the individuals' sensitive features by replacing them with the cluster's average values. Only the anonymized dataset will be shared with the learner who then uses it to train a model. We refer to Figure 1 for an illustration of this approach. Note that the learner never gets access to the individuals' sensitive features and so this approach is safe from re-identification attacks where the learner is given access to the pool of individuals' sensitive information (up to permutation) and may use the non-sensitive features to re-identify the

37th Conference on Neural Information Processing Systems (NeurIPS 2023).

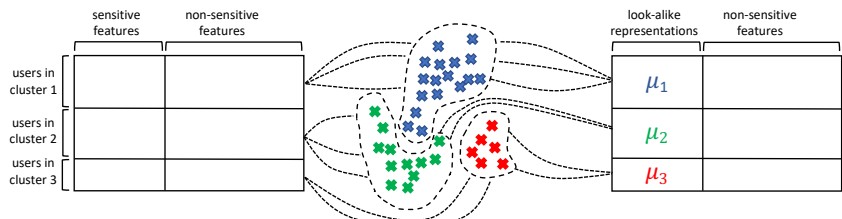

Figure 1: Schematic illustration of look-alike clustering on features data. Within each cluster, the sensitive features of users are replaced by a common look-alike representation (center of the cluster). In this example, $\mu_1, \mu_2, \mu_3$ represent the average of the sensitive features vectors for users in cluster 1, 2, 3.

users. Also note that since a common representation (average sensitive features) is used for all the users in a cluster, this approach offers $m$-anonymity provided that each cluster is of size at least $m$ (minimum size clustering).

Minimum size clustering has received an increased attention mainly as a tool for anonymization and when privacy considerations are in place [7, 2, 3]. A particular application is for providing anonymity for user targeting in online advertising with the goal of replacing the use of third-party cookies with a more privacy-respecting entity [10]. There are a variety of approximation algorithms for clustering with minimum size constraint [23, 9, 1, 24], as well as parallel and dynamic implementation [10].

In this paper, we focus on linear regression and derive a precise characterization of model generalization[1] using the look-alike clustering approach, in the so-called *proportional regime* where the size of training set grows in proportion to the number of parameters (which for the linear regression is equal to the number of features). The proportional regime has attracted a significant attention as overparametrized models have become greatly prevalent. It allows to understand the effect under/overparametrization in feature-rich models, providing insights to several intriguing phenomena, including double-descent behavior in the generalization error [22, 8, 14].

Our precise asymptotic theory allows us to demystify the effect of different factors on the model generalization under look-alike clustering, such as the role of cluster size, number of clusters, signal-to-noise ratio of the model as well as the strength of sensitive and non-sensitive features. A key tool in our analysis is a powerful extension of Gordon's Gaussian process inequality [13] known as the Convex Gaussian Minimax Theorem (CGMT), which was developed in [30] and has been used for studying different learning problems; see e.g, [29, 8, 15, 14, 16].

Initially, it might be presumed that look-alike clustering would hinder model generalization by suppressing sensitive features of individuals, suggesting a possible tradeoff between anonymity (privacy) and model performance. However, our analysis uncovers scenarios in which look-alike clustering actually enhances model generalization! We will develop further insights on these results by arguing that the proposed look-alike clustering can serve as a form of regularization, mitigating model overfitting and consequently improving the model generalization.

Before summarizing our key contributions in this paper, we conclude this section by discussing some of the recent work on the tradeoff between privacy and model generalization at large. An approach to study such potential tradeoff is via the lens of memorization. Modern deep neural networks, with remarkable generalization property, operate in the overparametrized regime where there are more tunable parameters than the number of training samples. Such overparametrized models tend to interpolate the training data and are known to fit well even random labels [34, 33]. Similar phenomenon has been observed in other models, such as random forest [4], Adaboost [25, 32], and kernel methods [5, 17]. Beyond label memorization, [6] studies setting where learning algorithms with near-optimal generalization must encode most of the information about the entire high-dimensional (and high-entropy) covariates of the training examples. Clearly, memorization of training data imposes significant privacy risks when this data contains sensitive personal information, and therefore these results hint to a potential trade-off between privacy protection and model generalization [27, 12, 20]. Lastly, [11] studies settings where data is sampled from a mixture of subpopulations, and shows that label memorization is *necessary* for achieving near-optimal generalization error, whenever the

---

[1]the ability of the model to generalize to new, unseen data from the same distribution as the training data

distribution of subpopulation frequencies is long-tailed. Intuitively, this corresponds to datasets with many small distinct subpopulations. In order to predict more accurately on a subpopulation from which only a very few examples are observed, the algorithm needs to memorize their labels.

## 1.1 Summary of contributions

We consider a linear regression setting for response variable $y$ given feature $\boldsymbol{x}$, and posit a Gaussian Mixture Model on the features to model the clustering structure on the samples. We focus on the high-dimensional asymptotic regime where the number of training samples $n$, the dimension of sensitive features $(p)$, and the dimension of non-sensitive features $(d - p)$ grow in proportion $(p/n \to \psi_p$ and $d/n \to \psi_d$, for some constants $0 < \psi_p \le \psi_d)$. Asymptotic analysis in this particular regime, characterized by a fixed sample size to feature size ratio, has recently garnered significant attention due to its relevance to the regime where modern neural networks operate. This analysis allows for the study of various intriguing phenomena related to both statistical properties (such as double-descent) and the tractability of optimizing the learning process in such networks [22, 8, 14], where the population analysis $n/d \to \infty$ fails to capture. Let $\mathcal{T}^n = \{(\boldsymbol{x}_i, y_i), i \in [n]\}$ denote the (unanonymized) training set and $\mathcal{T}_L^n$ be the set obtained after replacing the sensitive features with the look-alike representations of clusters. We denote by $\widehat{\boldsymbol{\theta}}$ and $\widehat{\boldsymbol{\theta}}_L$ the min-norm estimators fit to $\mathcal{T}^n$ and $\mathcal{T}_L^n$, respectively. Under this asymptotic setting:

- We provide a precise characterization of the generalization error of $\widehat{\boldsymbol{\theta}}$ and $\widehat{\boldsymbol{\theta}}_L$. Despite the randomness in data generating model, we show that in the high-dimensional asymptotic, the generalization errors of these estimators converge in probability to deterministic limits for which we provide explicit expressions.

- Our characterizations reveal several interesting facts about the generalization of the estimators:

  $(i)$ For the min-norm estimator $\widehat{\boldsymbol{\theta}}$ we observe significantly different behavior in the underparametrized regime $(\psi_d \le 1)$ than in the overparametrized regime $(\psi_d > 1)$. Note that in the underparametrized regime, the min-norm estimator coincides with the standard least squares estimator. For the look-alike estimator $\widehat{\boldsymbol{\theta}}_L$ our analysis identifies the underparametrized regime as $\psi_d - \psi_p \le 1$ and the overparametrized regime as $\psi_d - \psi_p > 1$.

  $(ii)$ In the underparametrized regime, our analysis shows that, somewhat surprisingly, the generalization error (for both estimators) does not depend on the number or size of the clusters, nor the scaling of the cluster centers.

  $(iii)$ In the overparametrized regime, our analysis provides a precise understanding of the role of different factors, including the number of clusters, energy of cluster centers, and the alignment of the model with the constellation of cluster centers, on the generalization error.

- Using our characterizations, we discuss settings where the look-alike estimator $\widehat{\boldsymbol{\theta}}_L$ has better generalization than its non-private counterpart $\widehat{\boldsymbol{\theta}}$. A relevant quantity that shows up in our analysis is the ratio of the norm of the model component on the sensitive features over the noise in the response, which we refer to as signal-to-noise ratio (SNR). Using our theory, we show that if SNR is below a certain threshold, then look-alike estimator $\widehat{\boldsymbol{\theta}}_L$ has lower generalization error than $\widehat{\boldsymbol{\theta}}$. This demonstrates scenarios where anonymizing sensitive features via look-alike clustering does 'not' hinder model generalization. We give an interpretation for this result, after Theorem 5.1, by arguing that at low-SNR, look-alike clustering acts as a regularization and mitigates overfitting, which consequently improves model generalization.

- In our analysis in the previous parts, we assume that the learner has access to the exact underlying clustering structure on the users, to disentangle the clustering estimation error from look-alike modeling. However, in practice the learner needs to estimate the clustering structure from data. In Section 3.2, we combine our analysis with a perturbation analysis to extend our results to the case of imperfect clustering estimation.

Due to space constraint, we refer to the supplementary material for an overview of our proof techniques as well as proof of theorems and technical lemmas.

## 2 Model

We consider a linear regression setting, where we are given $n$ i.i.d pairs $(\boldsymbol{x}_i, y_i)$, where the response $y_i$ is given by

$$y_i = \langle \boldsymbol{x}_i, \boldsymbol{\theta}_0 \rangle + \varepsilon_i, \quad \varepsilon_i \sim \mathsf{N}(0, \sigma^2). \tag{2.1}$$

We assume that there is a clustering structure on features $\boldsymbol{x}_i$, $i \in [n]$, independent from the responses. We model this structure via Gaussian-Mixture model.

**Gaussian-Mixture Model (GMM) on features.** Each example $\boldsymbol{x}$ belong to cluster $\ell \in [k]$, with probability $\pi_\ell$. We let $\boldsymbol{\pi} = [\pi, \pi_2, \ldots, \pi_k] \in \mathbb{R}^k$ with $\boldsymbol{\pi} \geq 0$ and $\mathbf{1}^\mathsf{T} \boldsymbol{\pi} = 1$. The cluster conditional distribution of an example $\boldsymbol{x}$ in cluster $\ell$ follows an isotropic Gaussian with mean $\boldsymbol{\mu}_\ell \in \mathbb{R}^d$, namely

$$\boldsymbol{x} = \boldsymbol{\mu}_\ell + \boldsymbol{z}, \quad \boldsymbol{z} \sim \mathsf{N}(\mathbf{0}, \tau^2 \boldsymbol{I}). \tag{2.2}$$

By scaling the model (2.1), without loss of generality we assume $\tau = 1$. Writing in the matrix form, we let

$$\boldsymbol{X} = [\boldsymbol{x}_1 | \boldsymbol{x}_2 | \ldots | \boldsymbol{x}_n] \in \mathbb{R}^{d \times n}, \quad \boldsymbol{y} = (y_1, \ldots, y_n) \in \mathbb{R}^n, \quad \boldsymbol{M} = [\boldsymbol{\mu}_1 | \boldsymbol{\mu}_2 | \ldots | \boldsymbol{\mu}_k] \in \mathbb{R}^{d \times k}. \tag{2.3}$$

It is also convenient to encode the cluster membership as one-hot encoded vectors $\boldsymbol{\lambda}_i \in \mathbb{R}^k$, where $\boldsymbol{\lambda}_i$ is one at entry $\ell$ (with $\ell$ being the cluster of example $\boldsymbol{x}_i$) and zero everywhere else. The GMM can then be written as

$$\boldsymbol{X} = \boldsymbol{M}\boldsymbol{\Lambda} + \boldsymbol{Z}, \tag{2.4}$$

with $\boldsymbol{Z} \in \mathbb{R}^{d \times n}$ is a Gaussian matrix with i.i.d $\mathsf{N}(0,1)$ entries, and $\boldsymbol{\Lambda} \in \mathbb{R}^{k \times n}$ is the matrix obtained by stacking vectors $\boldsymbol{\lambda}_i$ as its column.

**Sensitive and non-sensitive features.** We assume that some of the features are sensitive for which we have some reservation to share with the learner and some non-sensitive features. Without loss of generality, we write it as $\boldsymbol{x} = (\boldsymbol{x}_\mathrm{s}, \boldsymbol{x}_\mathrm{ns})$, where $\boldsymbol{x}_\mathrm{s} \in \mathbb{R}^p$ representing the sensitive features and $\boldsymbol{x}_\mathrm{ns} \in \mathbb{R}^{d-p}$ representing the non-sensitive features. We also decompose the model $\boldsymbol{\theta}_0$ (2.1) as $\boldsymbol{\theta}_0 = (\boldsymbol{\theta}_{0,\mathrm{s}}, \boldsymbol{\theta}_{0,\mathrm{ns}})$ with $\boldsymbol{\theta}_{0,\mathrm{s}} \in \mathbb{R}^p$ and $\boldsymbol{\theta}_{0,\mathrm{ns}} \in \mathbb{R}^{d-p}$. Likewise, the cluster mean vector $\boldsymbol{\mu}$ is decomposed as $\boldsymbol{\mu} = (\boldsymbol{\mu}_\mathrm{s}, \boldsymbol{\mu}_\mathrm{ns})$. The idea of look-alike clustering is to replace the sensitive features of an example $\boldsymbol{x}_\mathrm{s}$ with the center of its cluster $\boldsymbol{\mu}_\mathrm{s}$. This way, if each cluster is of size at least $m$, then look-alike clustering offers $m$-anonymity.

Our goal in this paper is to precisely characterize the effect of look-alike clustering on model generalization. We focus on the high-dimensional asymptotic regime, where the number of training data $n$, and features sizes $d, p$ grow in proportion.

We formalize the high-dimensional asymptotic setting in the assumption below:

**Assumption 1** *We assume that the number of clusters $k$ is fixed and focus on the asymptotic regime where $n, d, p \to \infty$ at a fixed ratio $d/n \to \psi_d$ and $p/n \to \psi_p$.*

To study the generalization of a model $\boldsymbol{\theta}$ (performance on unseen data) via the *out-of-sample prediction risk* defined as $\mathrm{Risk}(\boldsymbol{\theta}) := \mathbb{E}[(y - \boldsymbol{x}^\mathsf{T} \boldsymbol{\theta})^2]$, where $(y, \boldsymbol{x})$ is generated according to (2.1). Our next lemma characterizes the risk when the feature $\boldsymbol{x}$ is drawn from GMM.

**Lemma 2.1** *Under the linear response model* (2.1) *and a GMM for features $\boldsymbol{x}$, the out-of-sample prediction risk of a model $\boldsymbol{\theta}$ is given by*

$$\mathrm{Risk}(\boldsymbol{\theta}) = \sigma^2 + \|\boldsymbol{\theta}_0 - \boldsymbol{\theta}\|_{\ell_2}^2 + (\boldsymbol{\theta}_0 - \boldsymbol{\theta})^\mathsf{T} \boldsymbol{M} diag(\boldsymbol{\pi}) \boldsymbol{M}^\mathsf{T} (\boldsymbol{\theta}_0 - \boldsymbol{\theta}).$$

The proof of Lemma 2.1 is deferred to the supplementary.

## 3 Main results

Consider the minimum $\ell_2$ norm (min-norm) least squares regression estimator of $\boldsymbol{y}$ on $\boldsymbol{X}$ defined by

$$\widehat{\boldsymbol{\theta}} = (\boldsymbol{X}\boldsymbol{X}^\mathsf{T})^\dagger \boldsymbol{X}\boldsymbol{y}, \tag{3.1}$$

where $(\boldsymbol{X}\boldsymbol{X}^{\mathsf{T}})^{\dagger}$ denotes the Moore-Penrose pseudoinverse of $\boldsymbol{X}\boldsymbol{X}^{\mathsf{T}}$. This estimator can also be formulated as

$$\widehat{\boldsymbol{\theta}} := \arg\min\left\{\|\boldsymbol{\theta}\|_{\ell_2} : \boldsymbol{\theta} \text{ minimizes } \|\boldsymbol{y} - \boldsymbol{X}^{\mathsf{T}}\boldsymbol{\theta}\|_{\ell_2}\right\}.$$

We also define the "look-alike estimator" denoted by $\widehat{\boldsymbol{\theta}}_L$, where the sensitive features are first anonymized via look-like modeling, and then the min-norm estimator is computed based on the resulting features. Specifically the sensitive feature $\boldsymbol{x}_s$ of each sample is replaced by the center of its cluster. In our notation, writing $\boldsymbol{X}^{\mathsf{T}} = [\boldsymbol{X}_s^{\mathsf{T}}, \boldsymbol{X}_{ns}^{\mathsf{T}}]$, we define $\boldsymbol{X}_L^{\mathsf{T}} = [(\boldsymbol{M}_s\boldsymbol{\Lambda})^{\mathsf{T}}, \boldsymbol{X}_{ns}^{\mathsf{T}}]$ the features matrix obtained after look-alike modeling on the sensitive features. The look-alike estimator is then given by

$$\widehat{\boldsymbol{\theta}}_L = (\boldsymbol{X}_L\boldsymbol{X}_L^{\mathsf{T}})^{\dagger}\boldsymbol{X}_L\boldsymbol{y}, \tag{3.2}$$

Our main result is to provide a precise characterization of the risk of look-alike estimator $\widehat{\boldsymbol{\theta}}_L$ as well as $\widehat{\boldsymbol{\theta}}$ (non-look-alike) in the asymptotic regime, as described in Assumption 1. We then discuss regimes where look-alike clustering offers better generalization.

As our analysis shows there are two majorly different setting in the behavior of the look-alike estimator: $(i)$ $\psi_d - \psi_p \leq 1$, i.e., the sample size $n$ is asymptotically larger than $d - p$, the number of non-sensitive features (referred to as *underparametrized* asymptotics); $(ii)$ $\psi_d - \psi_p \geq 1$, which is referred to as *overparametrized* asymptotics.

Our first theorem is on the risk of look-alike estimator in the underparametrized setting. To present our result, we consider the following singular value decomposition for $\boldsymbol{M}_s$, the matrix of cluster centers restricted to sensitive features:

$$\boldsymbol{M}_s = \boldsymbol{U}_s\boldsymbol{\Sigma}_s\boldsymbol{V}_s^{\mathsf{T}}, \quad \boldsymbol{U}_s \in \mathbb{R}^{p \times r}, \boldsymbol{\Sigma}_s \in \mathbb{R}^{r \times r}, \boldsymbol{V}_s \in \mathbb{R}^{k \times r}, \quad \text{with } r = \text{rank}(\boldsymbol{M}_s) \leq k.$$

**Theorem 3.1** *(**Look-alike estimator, underparametrized regime***) Consider the linear response model (2.1), where the features are coming from the GMM (2.4). Also assume that $\|\boldsymbol{\theta}_{0,s}\| = r_s$ and $\|\boldsymbol{U}_s^{\mathsf{T}}\boldsymbol{\theta}_{0,s}\| = \sqrt{\rho}r_s$, for all $n, p$. Under Assumption 1 with $\psi_d - \psi_p \leq 1$, the out-of-sample prediction risk of look-alike estimator $\widehat{\boldsymbol{\theta}}_L$, defined by (3.2), converges in probability,*

$$\text{Risk}(\widehat{\boldsymbol{\theta}}_L) \xrightarrow{\mathcal{P}} \frac{\sigma^2 + r_s^2}{1 - (\psi_d - \psi_p)} - \rho r_s^2.$$

There are several intriguing observations about this result. In the underparametrized regime:

1. The risk depends on $\boldsymbol{\theta}_{0,s}$ (model component on the sensitive features), only through the norms $\|\boldsymbol{\theta}_{0,s}\| = r_s$ and $\|\boldsymbol{U}_s^{\mathsf{T}}\boldsymbol{\theta}_{0,s}\| = \sqrt{\rho}r_s$. Note that $\|\boldsymbol{U}_s^{\mathsf{T}}\boldsymbol{\theta}_{0,s}\|$ measures the alignment of the model with the left singular vectors of the cluster centers.

2. The cluster structure on the non-sensitive features plays no role in the risk, nor does $\boldsymbol{\theta}_{0,ns}$ the model component corresponding to the non-sensitive features.

3. The cluster prior probabilities $\boldsymbol{\pi}$ does not impact the risk.

We next proceed to the overparametrized setting. For technical convenience, we make some simplifying assumption, however, we believe a similar derivation can be obtained for the general case, albeit with a more involved analysis.

**Assumption 2** *Suppose that there is no cluster structure on the non-sensitive features ($\boldsymbol{M}_{ns} = \boldsymbol{0}$). Also, assume orthogonal, equal energy centers for the clusters on the sensitive features ($\boldsymbol{M}_s = \mu\boldsymbol{U}_s$ with $\boldsymbol{U}_s^{\mathsf{T}}\boldsymbol{U}_s = \boldsymbol{I}_k$).*

Our next theorem characterizes the risk of look-alike estimator in the underparametrized regime.

**Theorem 3.2** *(**Look-alike estimator, overparametrized regime***) Consider the linear response model (2.1), where the features are coming from the GMM (2.4). Also assume that $\|\boldsymbol{\theta}_{0,s}\| = r_s$, $\|\boldsymbol{\theta}_{0,ns}\| = r_{ns}$ and $\|\boldsymbol{U}_s^{\mathsf{T}}\boldsymbol{\theta}_{0,s}\| = \sqrt{\rho}r_s$, for all $n, p, d$. Under Assumption 1 with $\psi_d - \psi_p \geq 1$, and Assumption 2, the out-of-sample prediction risk of look-alike estimator $\widehat{\boldsymbol{\theta}}_L$, defined by (3.2), converges in probability,*

$$\text{Risk}(\widehat{\boldsymbol{\theta}}_L) \xrightarrow{\mathcal{P}} \sigma^2 + (1 - \rho)r_s^2 + \gamma_0^2 + \boldsymbol{\alpha}^{\mathsf{T}}(\boldsymbol{I} + \mu^2 diag(\boldsymbol{\pi}))\boldsymbol{\alpha}, \tag{3.3}$$

where $\boldsymbol{\pi} = (\pi_1, \ldots, \pi_k)$ encodes the cluster priors and $\gamma_0$ and $\boldsymbol{\alpha} \in \mathbb{R}^k$ are given by the following relations:

$$\boldsymbol{\alpha} = \left(\boldsymbol{I} + \frac{\mu^2 diag(\boldsymbol{\pi})}{\psi_d - \psi_p - 1}\right)^{-1} \boldsymbol{U}_{\mathrm{s}}^\top \boldsymbol{\theta}_{0,\mathrm{s}},$$

$$\gamma_0^2 = \frac{1}{\psi_d - \psi_p - 1}\left(\sigma^2 + r_{\mathrm{s}}^2 + \mu^2 \boldsymbol{\alpha}^\top diag(\boldsymbol{\pi})\boldsymbol{\alpha}\right) + \left(1 - \frac{1}{\psi_d - \psi_p}\right) r_{\mathrm{ns}}^2.$$

As discussed in the introduction, one of the focal interest in this work is to understand cases where look-alike modeling improves generalization. In Section 5 we discuss this by comparing the look-alike estimator $\widehat{\boldsymbol{\theta}}_L$ with the min-norm estimator $\widehat{\boldsymbol{\theta}}$, given by (3.1) which utilizes the full information on the sensitive features. In order to do that, we next derive a precise characterization of the risk of $\widehat{\boldsymbol{\theta}}$ in the asymptotic setting.

**Theorem 3.3** (**min-norm estimator with no look-alike clustering**) *Consider the linear response model* (2.1), *where the features are coming from the GMM* (2.4). *Under Assumption 1, the followings hold for the min-norm estimator $\widehat{\boldsymbol{\theta}}$ given by* (3.1):

*(a)* *(underparametrized setting) If $\psi_d \leq 1$, we have*

$$\mathrm{Risk}(\widehat{\boldsymbol{\theta}}) \overset{\mathcal{P}}{\to} \frac{\sigma^2}{1 - \psi_d}.$$

*(b)* *(overparametrized setting) If $\psi_d \geq 1$, under Assumption 2, the prediction risk of $\widehat{\boldsymbol{\theta}}$ converges in probability*

$$\mathrm{Risk}(\widehat{\boldsymbol{\theta}}) \overset{\mathcal{P}}{\to} \sigma^2 + \tilde{\gamma}_0^2 + \tilde{\boldsymbol{\alpha}}^\top(\boldsymbol{I} + \mu^2 diag(\boldsymbol{\pi}))\tilde{\boldsymbol{\alpha}}, \tag{3.4}$$

*where $\tilde{\gamma}_0$ and $\tilde{\boldsymbol{\alpha}}$ are given by the following relations:*

$$\tilde{\boldsymbol{\alpha}} = \left(\boldsymbol{I} + \frac{\boldsymbol{I} + \mu^2 diag(\boldsymbol{\pi})}{\psi_d - 1}\right)^{-1} \boldsymbol{U}_{\mathrm{s}}^\top \boldsymbol{\theta}_{0,\mathrm{s}},$$

$$\tilde{\gamma}_0^2 = \frac{1}{\psi_d - 1}\left(\sigma^2 + \tilde{\boldsymbol{\alpha}}^\top(\boldsymbol{I} + \mu^2 diag(\boldsymbol{\pi}))\tilde{\boldsymbol{\alpha}}\right) + \left(1 - \frac{1}{\psi_d}\right)\left((1 - \rho)r_{\mathrm{s}}^2 + r_{\mathrm{ns}}^2\right).$$

**Example 3.1** *(Balanced clusters) In the case of equal cluster prior $(\pi_1 = \ldots = \pi_k = 1/k)$, the risk characterization* (3.3) *depends on $\boldsymbol{\alpha}$ only through $\|\boldsymbol{\alpha}\|_{\ell_2}$ (and likewise, the risk* (3.4) *depends on $\tilde{\boldsymbol{\alpha}}$ only through its norm). This significantly simplifies these characterizations.*

## 3.2 Extension to imperfect clustering estimation

In our previous results, we assumed that the underlying cluster memberships of users are known to the learner, so we could concentrate our analysis on the impact of training using anonymous cluster centers. However, in practice, clusters should be estimated from the features and thus includes an estimation error. In our next result, we combine our previous result with a perturbation analysis to bound the risk of the look-alike estimator based on estimated clusters.

Recall matrix $\boldsymbol{M} \in \mathbb{R}^{d \times k}$ from (2.3), whose columns are the cluster centers. Also, recall the matrix $\boldsymbol{\Lambda} \in \mathbb{R}^{k \times n}$ whose columns are the one-hot encoding of the cluster memberships. We let $\widetilde{\boldsymbol{M}}$ and $\widetilde{\boldsymbol{\Lambda}}$ indicate the estimated matrices, with the cluster estimation error rate $\delta_n := \frac{1}{\sqrt{n}}\|\boldsymbol{M}_s\boldsymbol{\Lambda} - \widetilde{\boldsymbol{M}}_s\widetilde{\boldsymbol{\Lambda}}\|_2$, where $\|\cdot\|_2$ indicates spectral norm. Note that only the cluster estimation error with respect to the sensitive features matters because in the look-alike modeling only those features are anononymized (replaced by the cluster centers).

**Proposition 3.4** *Let $\widetilde{\boldsymbol{X}}^\top := [(\widetilde{\boldsymbol{M}}_s\widetilde{\boldsymbol{\Lambda}})^\top, \boldsymbol{X}_{\mathrm{ns}}^\top]$ be the feature matrix after replacing the sensitive features with the estimated cluster centers of users. We also let $\widetilde{\boldsymbol{\theta}}_L = (\widetilde{\boldsymbol{X}}_L \widetilde{\boldsymbol{X}}_L^\top)^\dagger \widetilde{\boldsymbol{X}}_L \boldsymbol{y}$ be the look-alike estimator based on $\widetilde{\boldsymbol{X}}_L$. Note that $\widetilde{\boldsymbol{\theta}}_L$ is the counterpart of $\widehat{\boldsymbol{\theta}}_L$ given by* (3.2). *Define the cluster estimation error rate $\delta_n := \frac{1}{\sqrt{n}}\|\boldsymbol{M}_s\boldsymbol{\Lambda} - \widetilde{\boldsymbol{M}}_s\widetilde{\boldsymbol{\Lambda}}\|_2$, and suppose that either of the following conditions hold:*

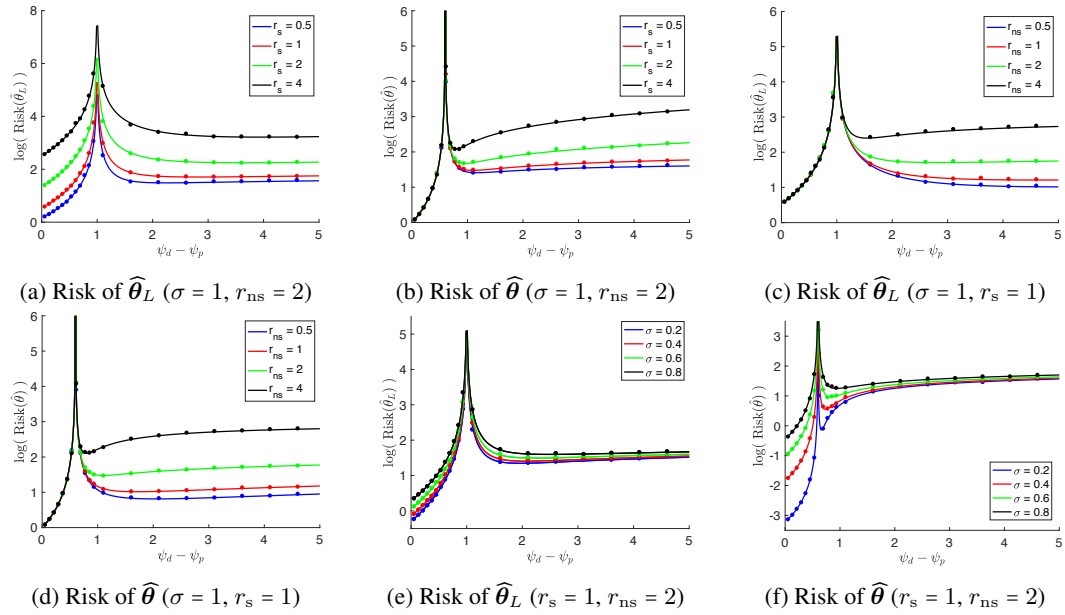

Figure 2: Validation of theoretical characterizations of the risks. Curves correspond to (asymptotic) analytical predictions, and dots to numerical simulations (averaged over 20 realizations). In all the plots, $d = 500$, $p = 200$, $\mu = 5$, $k = 3$, $\rho = 0.3$. Left panel corresponds to the risk of $\widehat{\boldsymbol{\theta}}_L$ and right panel corresponds to the risk of $\widehat{\boldsymbol{\theta}}$.

- *(i) $\psi_d - \psi_p < 0.5$ and $\delta < \sqrt{1 - (\psi_d - \psi_p)} - \sqrt{\psi_d - \psi_p}$.*
- *(ii) $\psi_d - \psi_p > 2$ and $\delta < \sqrt{\psi_d - \psi_p - 1} - 1$.*

*Then,*

$$\mathrm{Risk}(\widetilde{\boldsymbol{\theta}}_L) \le \mathrm{Risk}(\widehat{\boldsymbol{\theta}}_L) + C\delta\,,$$

*for some constant $C$ depending on the problem parameters.*

## 4  Numerical experiments

In this section, we validate our theory with numerical experiments. We consider GMM with $k$ clusters, where the centers of clusters are given by $\mu \boldsymbol{u}_\ell$, for $\ell \in [k]$, where $\boldsymbol{u}_\ell \in \mathbb{R}^d$ are of unit $\ell_2$-norm. Also the vectors $\boldsymbol{u}_\ell$ are non-zero only on the first $p$ entries, and their restriction to these entries form a random orthogonal constellation. Therefore, defining $\boldsymbol{U} = [\boldsymbol{u}_1, \ldots, \boldsymbol{u}_k]$, we have $\boldsymbol{U} = \begin{bmatrix} \boldsymbol{U}_{\mathrm{s}} \\ \boldsymbol{0} \end{bmatrix}$, with $\boldsymbol{U}_{\mathrm{s}}^\mathsf{T} \boldsymbol{U}_{\mathrm{s}} = \boldsymbol{I}_k$. In this setting there is no cluster structure on the non-sensitive features and the cluster centers on the sensitive features are orthogonal and of same norm.

Recall the decomposition of the model $\boldsymbol{\theta}_0 = (\boldsymbol{\theta}_{0,\mathrm{s}}, \boldsymbol{\theta}_{0,\mathrm{ns}})$, with $\boldsymbol{\theta}_0$ the true underlying model (2.1) and $\boldsymbol{\theta}_{0,\mathrm{s}}, \boldsymbol{\theta}_{0,\mathrm{ns}}$ the components corresponding to sensitive and non-sensitive features. We generate $\boldsymbol{\theta}_{0,\mathrm{ns}} \in \mathbb{R}^{d-p}$ to have i.i.d standard normal entries and then normalize it to have $\left\| \boldsymbol{\theta}_{0,\mathrm{ns}} \right\|_{\ell_2} = r_{\mathrm{ns}}$. For $\boldsymbol{\theta}_{0,\mathrm{s}}$, we generate $\boldsymbol{Z}_1, \boldsymbol{Z}_2 \sim \mathsf{N}(\boldsymbol{0}_p, \boldsymbol{I}_p)$, independently and let

$$\boldsymbol{\theta}_{0,\mathrm{s}} = r_{\mathrm{s}} \sqrt{\rho}\, \frac{\mathsf{P}_{\boldsymbol{U}_{\mathrm{s}}} \boldsymbol{z}_1}{\left\| \mathsf{P}_{\boldsymbol{U}_{\mathrm{s}}} \boldsymbol{z}_1 \right\|_{\ell_2}} + r_{\mathrm{s}} \sqrt{1 - \rho}\, \frac{(\boldsymbol{I} - \mathsf{P}_{\boldsymbol{U}_{\mathrm{s}}}) \boldsymbol{z}_2}{\left\| (\boldsymbol{I} - \mathsf{P}_{\boldsymbol{U}_{\mathrm{s}}}) \boldsymbol{z}_2 \right\|_{\ell_2}}\,,$$

where $\mathsf{P}_{\boldsymbol{U}_{\mathrm{s}}} := \boldsymbol{U}_{\mathrm{s}} \boldsymbol{U}_{\mathrm{s}}^\mathsf{T}$ is the projection onto column space of $\boldsymbol{U}_{\mathrm{s}}$. Therefore, $\left\| \boldsymbol{\theta}_{0,\mathrm{s}} \right\|_{\ell_2} = r_{\mathrm{s}}$ and $\left\| \boldsymbol{U}_{\mathrm{s}}^\mathsf{T} \boldsymbol{\theta}_{0,\mathrm{s}} \right\|_{\ell_2} = \sqrt{\rho} r_{\mathrm{s}}$. Note that $\rho$ quantifies the alignment of the model with the cluster centers, confined to the sensitive features.)

We will vary the values of $r_{\mathrm{s}}$ and $r_{\mathrm{ns}}$ in our experiments. We also consider the case of balanced clusters, so the cluster prior probabilities are all equal, $\pi_\ell = 1/k$, for $\ell \in [k]$. We set the number of

cluster $k = 3$, dimension of sensitive features $p = 200$ and the dimension of entire features vector $d = 500$. We also set $\mu = 5$ and $\rho = 0.3$. In our experiments, we vary the sample size $n$ and plot the risk of $\widehat{\boldsymbol{\theta}}_L$ and $\widehat{\boldsymbol{\theta}}$ versus $\psi_d - \psi_p = (d - p)/n$. We consider different settings, where we vary $r_{\mathrm{s}}$, $r_{\mathrm{ns}}$ and $\sigma$ (noise variance in model (2.1)). In Figure 2, we report the results. Curves correspond to our asymptotic theory and dots to the numerical simulations. (Each dot is obtained by averaging over 20 realizations of that configuration.) As we observe, in all scenarios our theoretical predictions are a perfect match to the empirical performance.

## 5    When does look-alike clustering improve generalization?

In Section 3, we provided a precise characterization of the risk of look-alike estimator $\widehat{\boldsymbol{\theta}}_L$ and its counterpart, the min-norm estimator $\widehat{\boldsymbol{\theta}}$ which utilizes the full information on the sensitive features. By virtue of these characterizations, we would like to understand regimes where the look-alike clustering helps with the model generalization, and the role of different problem parameters in achieving this improvement. Notably, since the look-alike estimator offers $m$-anonymity on the sensitive features (with $m$ the minimum size of clusters), our discussion here points out instances where data anonymization and model generalization are not in-conflict.

We define the gain of look-alike estimator as $\Delta := \mathrm{Risk}(\widehat{\boldsymbol{\theta}})/\mathrm{Risk}(\widehat{\boldsymbol{\theta}}_L)$ to indicate the gain obtained in generalization via look-alike clustering. For ease in presentation, we focus on the case of balanced clusters (equal priors $\pi_1 = \ldots = \pi_k = 1/k$), and consider three cases:

• **Case 1** ($\psi_d \leq 1$): In this case, both $\widehat{\boldsymbol{\theta}}_L$ and $\widehat{\boldsymbol{\theta}}$ are in the underparametrized regime and Theorems 3.1 and 3.3 (a) provide simple closed-form characterization of the risks of $\widehat{\boldsymbol{\theta}}_L$ and $\widehat{\boldsymbol{\theta}}$, by which we obtain

$$\Delta \xrightarrow{\mathcal{P}} \frac{(1 - \psi_d)^{-1}}{(1 + r_{\mathrm{s}}^2/\sigma^2)(1 - \psi_d + \psi_p)^{-1} - \rho r_{\mathrm{s}}^2/\sigma^2} \,.$$

Define the signal-to-noise ratio $\mathrm{SNR} = r_{\mathrm{s}}/\sigma$. Since $\rho \leq 1$, it is easy to see that $\Delta$ is decreasing in the SNR. In particular, as $\mathrm{SNR} \to 0$, we have $\Delta \to (1 - \psi_d + \psi_p)/(1 - \psi_d) > 1$, which means the look-alike estimator $\widehat{\boldsymbol{\theta}}_L$ achieves lower risk compared to $\widehat{\boldsymbol{\theta}}$. In Figure 3a we plot $\log(\Delta)$ versus SNR, for several values of $\psi_p$. Here we set $\psi_d = 0.9$ and $\rho = 0.3$. As we observe in low SNR, the look-alike estimator has lower risk. Specifically, for each curve there is a threshold for the SNR, below which $\log(\Delta) > 0$. Furthermore, this threshold increases with $\psi_p$, covering a larger range of SNR where $\widehat{\boldsymbol{\theta}}_L$ has better generalization.

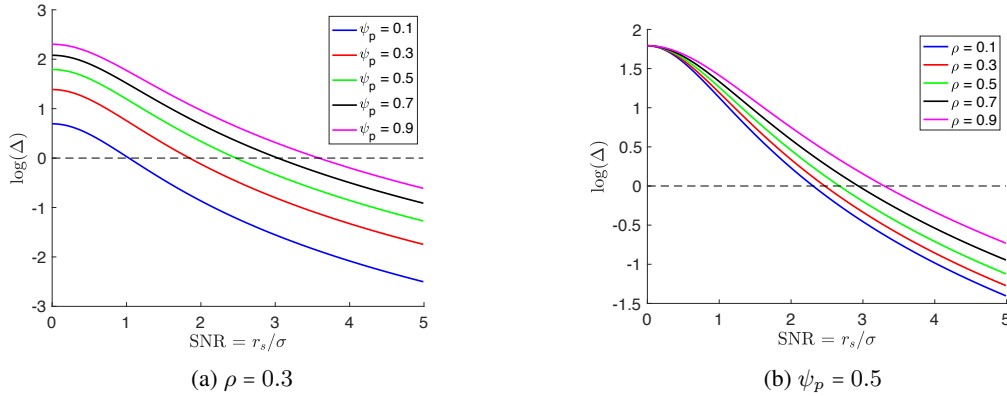

Figure 3: Behavior of gain $\Delta$ in the generalization of the look-alike estimator $\widehat{\boldsymbol{\theta}}_L$ over min-norm estimator $\widehat{\boldsymbol{\theta}}$ as we vary $\mathrm{SNR} = r_{\mathrm{s}}/\sigma$. Here, $\psi_d = 0.9$, $\sigma = 1$, and we are in the underparametrized regime for both $\widehat{\boldsymbol{\theta}}_L$ and $\widehat{\boldsymbol{\theta}}$.

In Figure 3b we report similar curves, where this time $\psi_p = 0.5$ and we consider several values of $\rho$. As we observe, at fixed SNR the gain $\Delta$ is increasing in $\rho$. This is expected since $\rho$ measures the alignment of the underlying model $\boldsymbol{\theta}_0$ with the (left eigenvectors of) cluster centers and so higher $\rho$ is to advantage of the look-alike estimator which uses the cluster centers instead of individuals' sensitive features.

• **Case 2** ($\psi_d \geq 1, \psi_d - \psi_p \leq 1$)**:** In this case, the look-alike estimator $\widehat{\boldsymbol{\theta}}_L$ is in the underparametrized regime, while the min-norm $\widehat{\boldsymbol{\theta}}$ is in the overparametrized regime. The following theorem uses the characterizations in Theorem 3.1 and and 3.3 (b), and shows that in the low SNR= $r_s/\sigma$, the look-alike estimator $\widehat{\boldsymbol{\theta}}_L$ has a positive gain. It further shows the monotonicity of the gain with respect to different problem parameters.

**Theorem 5.1** *Suppose that $\psi_d \geq 1$ and $\psi_d - \psi_p \leq 1$, and consider the case of equal cluster priors. The gain $\Delta$ is increasing in $r_{\mathrm{ns}}$ and $\rho$, and is decreasing in $\mu^2/k$. Furthermore, under the following condition*

$$\mathrm{SNR}^2 := \left(\frac{r_s}{\sigma}\right)^2 \leq \frac{1 + (\psi_d - 1)^{-1} - (1 - \psi_d + \psi_p)^{-1}}{(1 - \psi_d + \psi_p)^{-1} + \psi_d^{-1} - 1}, \tag{5.1}$$

*we have $\Delta \geq 1$, for all values of other parameters ($\mu, k, \rho, r_{\mathrm{ns}}$).*

**An interpretation based on regularization:** We next provide an argument to build further insight on the result of Theorem 5.1. Recall the data model (2.1), where substituting from (2.2) and decomposing over sensitive and non-sensitive features we arrive at

$$y = \langle \boldsymbol{x}_s, \boldsymbol{\theta}_s \rangle + \langle \boldsymbol{x}_{\mathrm{ns}}, \boldsymbol{\theta}_{\mathrm{ns}} \rangle + \varepsilon$$
$$= \langle \boldsymbol{\mu}_s, \boldsymbol{\theta}_s \rangle + \langle \boldsymbol{z}_s, \boldsymbol{\theta}_s \rangle + \langle \boldsymbol{x}_{\mathrm{ns}}, \boldsymbol{\theta}_{\mathrm{ns}} \rangle + \varepsilon .$$

Note that $\langle \boldsymbol{z}_s, \boldsymbol{\theta}_s \rangle \sim \mathsf{N}(0, \|\boldsymbol{\theta}_s\|^2)$. At low SNR, this term is of order of the noise term $\varepsilon \sim \mathsf{N}(0, \sigma^2)$. Recall that the look-alike clustering approach replaces the sensitive feature $\boldsymbol{x}_s$ by the cluster center $\boldsymbol{\mu}_s$, and therefore drops the term $\langle \boldsymbol{z}_s, \boldsymbol{\theta}_s \rangle$ from the model during the training process. In other words, look-alike clustering acts as a form of regularization which prevents overfitting to the noisy component $\langle \boldsymbol{z}_s, \boldsymbol{\theta}_s \rangle$, and this will help with the model generalization, together with anonymizing the sensitive features.

In Figure 4a we plot $\log(\Delta)$ versus $\mu$ for several values of $r_{\mathrm{ns}}$. Here, $\psi_d = 2$, $\psi_p = 1.7$, $\sigma = 1$, $r_s = 0.5$ and so condition (5.1) holds. As we observe $\log(\Delta)$ is positive, decreasing in $\mu$ and also at any fixed $\mu$, it is increasing in $r_{\mathrm{ns}}$, all of which are consistent with the Theorem 5.1. In Figure 4b, we plot similar curves where this time $r_{\mathrm{ns}} = 0.2$ and we try several values of $\rho$. As we see the look-alike estimator has larger gain $\Delta$ at larger values of $\rho$.

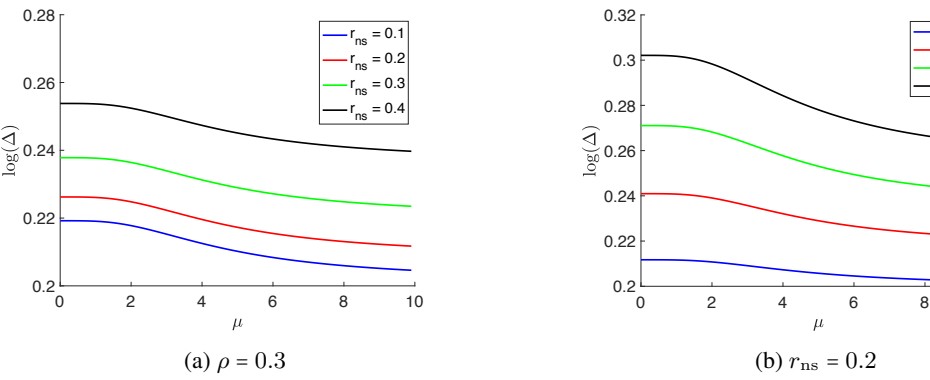

(a) $\rho = 0.3$           (b) $r_{\mathrm{ns}} = 0.2$

Figure 4: Behavior of gain $\Delta$ in the generalization of the look-alike estimator $\widehat{\boldsymbol{\theta}}_L$ over min-norm estimator $\widehat{\boldsymbol{\theta}}$ as we vary $\mu$ the energy of cluster centers.

• **Case 3** ($\psi_d - \psi_p \geq 1$)**:** In this case, both $\widehat{\boldsymbol{\theta}}_L$ and $\widehat{\boldsymbol{\theta}}$ are in the overparametrized regime. Let us first focus on $r_{\mathrm{ns}}$, the energy of the model on the non-sensitive features. Invoking the equations (3.3) and (3.4) and hiding the terms that do not depend on $r_{\mathrm{ns}}$ in constants $C_1, C_2$ we arrive at

$$\Delta \overset{(\mathcal{P})}{\to} \frac{C_1 + (1 - \frac{1}{\psi_d})r_{\mathrm{ns}}^2}{C_2 + (1 - \frac{1}{\psi_d - \psi_p})r_{\mathrm{ns}}^2} .$$

Therefore, $\lim_{r_{\mathrm{ns}} \to \infty} \Delta = (1 - \psi_d^{-1})/(1 - (\psi_d - \psi_p)^{-1}) > 1$, indicating a gain for the look-alike estimator over $\widehat{\boldsymbol{\theta}}$. In Figure 5a, we plot $\log(\Delta)$ versus $r_{\mathrm{ns}}$ for several values of $\psi_p$. As we observe, when $r_{\mathrm{ns}}$ is large enough we always have a gain, which is increasing in $\psi_p$.

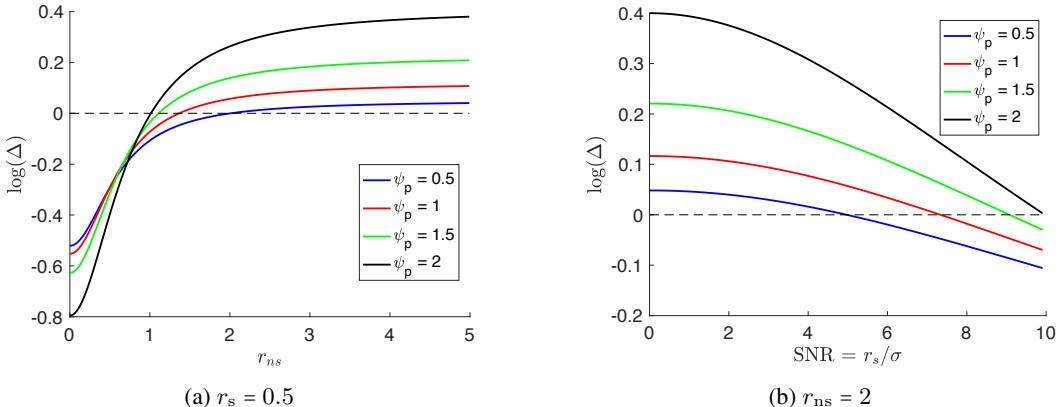

(a) $r_{\mathrm{s}} = 0.5$              (b) $r_{\mathrm{ns}} = 2$

Figure 5: Behavior of gain $\Delta$ versus $r_{\mathrm{ns}}$ and SNR:$=r_{\mathrm{s}}/\sigma$ for several values of $\psi_p$. Here, $\psi_d = 4$, $\sigma = 0.1$, $\rho = 0.3$, $\mu = 5$, $k = 5$. Here, we are in the overparametrized regime for both $\widehat{\boldsymbol{\theta}}_L$ and $\widehat{\boldsymbol{\theta}}$.

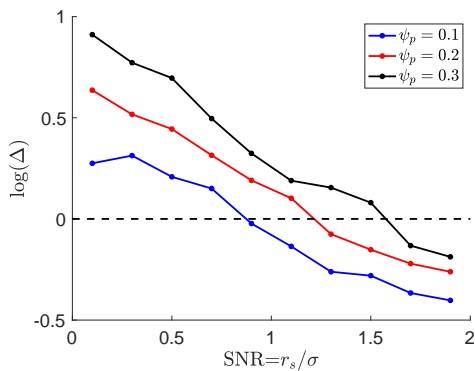

Figure 6: Behavior of gain $\Delta$ versus SNR for the nonlinear model described in Section 6. At small SNR, we observe a positive gain (lower risk of look-alike estimator $\widehat{\boldsymbol{\theta}}_L$ compared to $\widehat{\boldsymbol{\theta}}$).

We next consider the effect of SNR $= r_{\mathrm{s}}/\sigma$. In Figure 5b we $\log(\Delta)$ versus SNR, for several values of $\psi_p$. Similar to the underparametrized regime, we observe that in low SNR, the look-alike estimator has better generalization ($\log(\Delta) > 0$).

## 6 Beyond linear models

In previous section, we used our theory for linear models to show that at low SNR, look-alike modeling improves model generalization. We also provided an insight for this phenomenon by arguing that look-alike modeling acts as a form of regularization and avoids over-fitting at low SNR regime. In this section we show empirically that this phenomenon also extends to non-linear models.

Consider the following data generative model:

$$y \sim \mathrm{Binomial}(N, p_{\boldsymbol{x}}), \quad p_{\boldsymbol{x}} = \frac{1}{1 + \exp(-\langle \boldsymbol{x}, \boldsymbol{\theta}_0 \rangle + \varepsilon)},$$

where $\varepsilon \sim \mathsf{N}(0, \sigma^2)$. We construct the model $\boldsymbol{\theta}_0 = (\boldsymbol{\theta}_{0,\mathrm{s}}, \boldsymbol{\theta}_{0,\mathrm{ns}})$ similar to the setup in Section 4. We set $n = 200$, $d = 180$, $k = 3$, $\mu = 5$, $\sigma = 1$, $\rho = 0.3$, $r_{\mathrm{ns}} = 2$ and $N = 1000$ (number of trials in Binomial distribution). We vary SNR by changing $r_{\mathrm{s}}$ in the set $\{0.1, 0.3, \ldots, 1.9\}$. The estimators $\widehat{\boldsymbol{\theta}}$ and $\widehat{\boldsymbol{\theta}}_L$ are obtained by fitting a GLM with logit link function and binomial distribution. We compute the risks of $\widehat{\boldsymbol{\theta}}$ and $\widehat{\boldsymbol{\theta}}_L$ by averaging over a test set of size $50K$. In Figure 6, we plot the gain $\log(\Delta)$ versus $r_{\mathrm{s}}$ where each data point is by averaging over 50 different realizations of data. As we observe at low SNR, $\log(\Delta) > 0$ indicating that the look-alike estimator $\widehat{\boldsymbol{\theta}}_L$ obtains a lower risk than the min-norm estimator.

## Acknowledgement

This work is supported in part by the NSF CAREER Award DMS-1844481 and the NSF Award DMS-2311024.

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

# Supplementary Material

In this supplementary, we first provide an overview of our proof techniques in Appendix A and then in Appendix B provide the proofs of theorems and technical lemmas stated in the main paper.

## A    Overview of proof techniques

Our analysis of the generalization error is based on an extension of Gordon's Gaussian process inequality [13], called Convex-Gaussian Minimax Theorem (CGMT) [30]. Here, we outline the general steps of this framework and refer to the supplementary for complete details and derivations.

Consider the following two Gaussian processes:

$$
\begin{aligned}
\boldsymbol{X_{u,v}} &:= \boldsymbol{u}^\mathsf{T} \boldsymbol{G} \boldsymbol{v} + \psi(\boldsymbol{u}, \boldsymbol{v}) , \\
\boldsymbol{Y_{u,v}} &:= \|\boldsymbol{u}\|_{\ell_2} \boldsymbol{g}^\mathsf{T} \boldsymbol{v} + \|\boldsymbol{v}\|_{\ell_2} \boldsymbol{h}^\mathsf{T} \boldsymbol{u} + \psi(\boldsymbol{u}, \boldsymbol{v}) ,
\end{aligned}
$$

where $\boldsymbol{G} \in \mathbb{R}^{n \times d}$, $\boldsymbol{g} \in \mathbb{R}^n$ and $\boldsymbol{h} \in \mathbb{R}^d$, all have i.i.d standard normal entries. Further, $\psi : \mathbb{R}^d \times \mathbb{R}^n \to \mathbb{R}$ is a continuous function, which is convex in the first argument and concave in the second argument.

Given the above two processes, consider the following min-max optimization problems, which are respectively referred to as the *Primary Optimization (PO)* and the *Auxiliary Optimization (AO)* problems:

$$
\begin{aligned}
\Phi_{\mathrm{PO}}(\boldsymbol{G}) &:= \min_{\boldsymbol{u} \in \boldsymbol{S_u}} \max_{\boldsymbol{v} \in \boldsymbol{S_v}} \boldsymbol{X_{u,v}} , & \text{(A.1)} \\
\Phi_{\mathrm{AO}}(\boldsymbol{g}, \boldsymbol{h}) &:= \min_{\boldsymbol{u} \in \boldsymbol{S_u}} \max_{\boldsymbol{v} \in \boldsymbol{S_v}} \boldsymbol{Y_{u,v}} . & \text{(A.2)}
\end{aligned}
$$

The main result of CGMT is to connect the above two random optimization problems. As shown in [30](Theorem 3), if $\boldsymbol{S_u}$ and $\boldsymbol{S_v}$ are compact and convex then, for any $\lambda \in \mathbb{R}$ and $t > 0$,

$$
\mathbb{P}\left(|\Phi_{\mathrm{PO}}(\boldsymbol{G}) - \lambda| > t\right) \leq 2\mathbb{P}\left(|\Phi_{\mathrm{AO}}(\boldsymbol{g}, \boldsymbol{h}) - \lambda| > t\right) .
$$

An immediate corollary of this result (by choosing $\lambda = \mathbb{E}[\Phi_{\mathrm{AO}}(\boldsymbol{g}, \boldsymbol{h})]$) is that if the optimal cost of AO problem concentrates in probability, then the optimal cost of the corresponding PO problem also concentrates, in probability, around the same value. In addition, as shown in part (iii) of [30](Theorem 3), concentration of the optimal solution of the AO problem implies concentration of the optimal solution of the PO around the same value. Therefore, the two optimization are intimately connected and by analyzing the AO problem, which is substantially simpler, one can derive corresponding properties of the PO problem.

The CGMT framework has been used to infer statistical properties of estimators in certain high-dimensional asymptotic regime. The intermediate steps in the CGMT framework can be summarized as follows: First form an PO problem in the form of (A.1) and construct the corresponding AO problem. Second, derive the point-wise limit of the AO objective in terms of a convex-concave optimization problem, over only few scalar variables. This step is called 'scalarization'. Next, it is possible to establish uniform convergence of the scalarized AO to the (deterministic) min-max optimization problem using convexity conditions. Finally, by analyzing the latter deterministic problem, one can derive the desired asymptotic characterizations.

Of course implementing the above steps involved problem-specific intricate calculations. Our proofs of Theorems 3.1, 3.2, 3.3 in the supplementary follow this general strategy.

# B  Proof of theorems and technical lemmas

## B.1  Proof of Lemma 2.1

By substituting for $y$ from (2.1) in the definition of risk we obtain

$$
\begin{aligned}
\mathrm{Risk}(\boldsymbol{\theta}) &= \mathbb{E}[(y - \boldsymbol{x}^{\mathsf{T}}\boldsymbol{\theta})^2] \\
&= \mathbb{E}[(\boldsymbol{x}^{\mathsf{T}}(\boldsymbol{\theta}_0 - \boldsymbol{\theta}))^2] + \mathbb{E}[\varepsilon^2] \\
&\overset{(a)}{=} \sum_{\ell \in [k]} \pi_\ell \, \mathbb{E}[((\boldsymbol{\mu}_\ell + \boldsymbol{z})^{\mathsf{T}}(\boldsymbol{\theta}_0 - \boldsymbol{\theta}))^2] + \mathbb{E}[\varepsilon^2] \\
&= \sum_{\ell \in [k]} \pi_\ell \, \mathbb{E}[(\boldsymbol{\mu}_\ell^{\mathsf{T}}(\boldsymbol{\theta}_0 - \boldsymbol{\theta}))^2] + \sum_{\ell \in [k]} \pi_\ell \, \|\boldsymbol{\theta} - \boldsymbol{\theta}_0\|_{\ell_2}^2 + \sigma^2 \\
&\overset{(b)}{=} (\boldsymbol{\theta} - \boldsymbol{\theta}_0)^{\mathsf{T}} M \mathrm{diag}(\boldsymbol{\pi}) M^{\mathsf{T}} (\boldsymbol{\theta}_0 - \boldsymbol{\theta}) + \|\boldsymbol{\theta}_0 - \boldsymbol{\theta}\|_{\ell_2} + \sigma^2 \,,
\end{aligned}
$$

where $(a)$ follows from the Gaussian-Mixture model (2.2) and $(b)$ holds since $\sum_{\ell \in [k]} \pi_\ell = 1$.

## B.2  Proof of Theorem 3.1 and Theorem 3.2

Recall that the look-alike estimator is defined as the min-norm estimator over the feature matrix $X_L$, where the look-alike representations are used instead of individual sensitive features; see (3.2).

To analyze risk of $\widehat{\boldsymbol{\theta}}_L$, we consider the ridge regression estimator given by

$$
\widehat{\boldsymbol{\theta}}_\lambda = \arg\min_{\boldsymbol{\theta}} \frac{1}{2n} \left\| \boldsymbol{y} - X_L^{\mathsf{T}} \boldsymbol{\theta} \right\|_{\ell_2}^2 + \lambda \|\boldsymbol{\theta}\|_{\ell_2}^2 \,.
$$

The minimum-norm estimator is given by $\widehat{\boldsymbol{\theta}}_L = \lim_{\lambda \to 0^+} \widehat{\boldsymbol{\theta}}_\lambda$.

We follow the CGMT framework explained in Section A. Recall that

$$
X_L = \begin{bmatrix} M_{\mathrm{s}} \Lambda \\ M_{\mathrm{ns}} \Lambda + Z_{\mathrm{ns}} \end{bmatrix},
$$

and therefore by substituting for $\boldsymbol{y}$, $X$, and $X_L$, we get

$$
\begin{aligned}
\frac{1}{2n} \left\| \boldsymbol{y} - X_L^{\mathsf{T}} \boldsymbol{\theta} \right\|_{\ell_2}^2 &= \frac{1}{2n} \left\| \varepsilon + X^{\mathsf{T}} \boldsymbol{\theta}_0 - X_L^{\mathsf{T}} \boldsymbol{\theta} \right\|_{\ell_2}^2 \\
&= \frac{1}{2n} \left\| \varepsilon + \Lambda^{\mathsf{T}} M_{\mathrm{s}}^{\mathsf{T}} (\boldsymbol{\theta}_{0,\mathrm{s}} - \boldsymbol{\theta}_{\mathrm{s}}) + Z_{\mathrm{s}}^{\mathsf{T}} \boldsymbol{\theta}_{0,\mathrm{s}} + (\Lambda^{\mathsf{T}} M_{\mathrm{ns}}^{\mathsf{T}} + Z_{ns}^{\mathsf{T}})(\boldsymbol{\theta}_{0,\mathrm{ns}} - \boldsymbol{\theta}_{\mathrm{ns}}) \right\|_{\ell_2}^2 \,.
\end{aligned}
$$

We define the primary optimization loss as follows:

$$
\mathcal{L}_{PO}(\boldsymbol{\theta}_{\mathrm{s}}, \boldsymbol{\theta}_{\mathrm{ns}}) := \frac{1}{2n} \left\| \varepsilon + \Lambda^{\mathsf{T}} M_{\mathrm{s}}^{\mathsf{T}} (\boldsymbol{\theta}_{0,\mathrm{s}} - \boldsymbol{\theta}_{\mathrm{s}}) + Z_{\mathrm{s}}^{\mathsf{T}} \boldsymbol{\theta}_{0,\mathrm{s}} + (\Lambda^{\mathsf{T}} M_{\mathrm{ns}}^{\mathsf{T}} + Z_{ns}^{\mathsf{T}})(\boldsymbol{\theta}_{0,\mathrm{ns}} - \boldsymbol{\theta}_{\mathrm{ns}}) \right\|_{\ell_2}^2 + \lambda \|\boldsymbol{\theta}_{\mathrm{s}}\|_{\ell_2}^2 + \lambda \|\boldsymbol{\theta}_{\mathrm{ns}}\|_{\ell_2}^2
$$

We continue by deriving the auxiliary optimization (AO) problem. By duality, we have

$$
\begin{aligned}
\mathcal{L}_{PO}(\boldsymbol{\theta}_{\mathrm{s}}, \boldsymbol{\theta}_{\mathrm{ns}}) = \max_{\boldsymbol{v}} \frac{1}{n} &\left( \boldsymbol{v}^{\mathsf{T}} \varepsilon + \boldsymbol{v}^{\mathsf{T}} \Lambda^{\mathsf{T}} M_{\mathrm{s}}^{\mathsf{T}} (\boldsymbol{\theta}_{0,\mathrm{s}} - \boldsymbol{\theta}_{\mathrm{s}}) + \boldsymbol{v}^{\mathsf{T}} Z_{\mathrm{s}}^{\mathsf{T}} \boldsymbol{\theta}_{0,\mathrm{s}} + \boldsymbol{v}^{\mathsf{T}} (\Lambda^{\mathsf{T}} M_{\mathrm{ns}}^{\mathsf{T}} + Z_{ns}^{\mathsf{T}})(\boldsymbol{\theta}_{0,\mathrm{ns}} - \boldsymbol{\theta}_{\mathrm{ns}}) - \frac{\|\boldsymbol{v}\|_{\ell_2}^2}{2} \right) \\
&+ \lambda \|\boldsymbol{\theta}_{\mathrm{s}}\|_{\ell_2}^2 + \lambda \|\boldsymbol{\theta}_{\mathrm{ns}}\|_{\ell_2}^2
\end{aligned}
$$

Note that the above is jointly convex in $(\boldsymbol{\theta}_{\mathrm{s}}, \boldsymbol{\theta}_{\mathrm{ns}})$ and concave in $\boldsymbol{v}$, and the Gaussian matrix $Z$ is independent of everything else. Therefore, the AO problem reads:

$$
\begin{aligned}
\mathcal{L}_{AO}(\boldsymbol{\theta}_{\mathrm{s}}, \boldsymbol{\theta}_{\mathrm{ns}}) = \max_{\boldsymbol{v}} \frac{1}{n} \Big( &\boldsymbol{v}^{\mathsf{T}} \varepsilon + \boldsymbol{v}^{\mathsf{T}} \Lambda^{\mathsf{T}} M_{\mathrm{s}}^{\mathsf{T}} (\boldsymbol{\theta}_{0,\mathrm{s}} - \boldsymbol{\theta}_{\mathrm{s}}) \\
&+ \|\boldsymbol{\theta}_{0,\mathrm{s}}\|_{\ell_2} \boldsymbol{g}_{\mathrm{s}}^{\mathsf{T}} \boldsymbol{v} + \|\boldsymbol{v}\|_{\ell_2} \boldsymbol{h}_{\mathrm{s}}^{\mathsf{T}} \boldsymbol{\theta}_{0,\mathrm{s}} \\
&+ \|\boldsymbol{\theta}_{0,\mathrm{ns}} - \boldsymbol{\theta}_{\mathrm{ns}}\|_{\ell_2} \boldsymbol{g}_{\mathrm{ns}}^{\mathsf{T}} \boldsymbol{v} + \|\boldsymbol{v}\|_{\ell_2} \boldsymbol{h}_{\mathrm{ns}}^{\mathsf{T}} (\boldsymbol{\theta}_{0,\mathrm{ns}} - \boldsymbol{\theta}_{\mathrm{ns}}) \\
&+ \boldsymbol{v}^{\mathsf{T}} \Lambda^{\mathsf{T}} M_{\mathrm{ns}}^{\mathsf{T}} (\boldsymbol{\theta}_{0,\mathrm{ns}} - \boldsymbol{\theta}_{\mathrm{ns}}) - \frac{\|\boldsymbol{v}\|_{\ell_2}^2}{2} \Big) + \lambda \|\boldsymbol{\theta}_{\mathrm{s}}\|_{\ell_2}^2 + \lambda \|\boldsymbol{\theta}_{\mathrm{ns}}\|_{\ell_2}^2 \,,
\end{aligned}
$$

where $\boldsymbol{g}_\mathrm{s}, \boldsymbol{g}_\mathrm{ns} \in \mathbb{R}^n$ and $\boldsymbol{h}_\mathrm{s} \in \mathbb{R}^p$, $\boldsymbol{h}_\mathrm{ns} \in \mathbb{R}^{d-p}$ are independent Gaussian random vectors with i.i.d $\mathsf{N}(0,1)$ entries.

We next fix norm of $\|\boldsymbol{v}\|_{\ell_2} = \beta$, and maximize over its direction to obtain

$$
\begin{aligned}
\mathcal{L}_{AO}(\boldsymbol{\theta}_\mathrm{s}, \boldsymbol{\theta}_\mathrm{ns}) &= \max_{\beta \geq 0} \frac{1}{n}\Big( \beta \left\| \boldsymbol{\varepsilon} + \boldsymbol{\Lambda}^\mathsf{T} \boldsymbol{M}_\mathrm{s}^\mathsf{T}(\boldsymbol{\theta}_{0,\mathrm{s}} - \boldsymbol{\theta}_\mathrm{s}) + \|\boldsymbol{\theta}_{0,\mathrm{s}}\|_{\ell_2} \boldsymbol{g}_\mathrm{s} + \|\boldsymbol{\theta}_{0,\mathrm{ns}} - \boldsymbol{\theta}_\mathrm{ns}\|_{\ell_2} \boldsymbol{g}_\mathrm{ns} + \boldsymbol{\Lambda}^\mathsf{T} \boldsymbol{M}_\mathrm{ns}^\mathsf{T}(\boldsymbol{\theta}_{0,\mathrm{ns}} - \boldsymbol{\theta}_\mathrm{ns}) \right\|_{\ell_2} \\
&\qquad + \beta \boldsymbol{h}_\mathrm{s}^\mathsf{T} \boldsymbol{\theta}_{0,\mathrm{s}} + \beta \boldsymbol{h}_\mathrm{ns}^\mathsf{T}(\boldsymbol{\theta}_{0,\mathrm{ns}} - \boldsymbol{\theta}_\mathrm{ns}) - \frac{\beta^2}{2} \Big) + \lambda \|\boldsymbol{\theta}_\mathrm{s}\|_{\ell_2}^2 + \lambda \|\boldsymbol{\theta}_\mathrm{ns}\|_{\ell_2}^2 \\
&= \max_{\beta \geq 0} \frac{1}{n}\Big( \beta \left\| \boldsymbol{\varepsilon} + \boldsymbol{\Lambda}^\mathsf{T} \boldsymbol{M}_\mathrm{s}^\mathsf{T}(\boldsymbol{\theta}_{0,\mathrm{s}} - \boldsymbol{\theta}_\mathrm{s}) + \boldsymbol{\Lambda}^\mathsf{T} \boldsymbol{M}_\mathrm{ns}^\mathsf{T}(\boldsymbol{\theta}_{0,\mathrm{ns}} - \boldsymbol{\theta}_\mathrm{ns}) + \sqrt{\|\boldsymbol{\theta}_{0,\mathrm{s}}\|_{\ell_2}^2 + \|\boldsymbol{\theta}_{0,\mathrm{ns}} - \boldsymbol{\theta}_\mathrm{ns}\|_{\ell_2}^2}\, \boldsymbol{g} \right\|_{\ell_2} \\
&\qquad + \beta \boldsymbol{h}_\mathrm{s}^\mathsf{T} \boldsymbol{\theta}_{0,\mathrm{s}} + \beta \boldsymbol{h}_\mathrm{ns}^\mathsf{T}(\boldsymbol{\theta}_{0,\mathrm{ns}} - \boldsymbol{\theta}_\mathrm{ns}) - \frac{\beta^2}{2} \Big) + \lambda \|\boldsymbol{\theta}_\mathrm{s}\|_{\ell_2}^2 + \lambda \|\boldsymbol{\theta}_\mathrm{ns}\|_{\ell_2}^2 \,,
\end{aligned}
$$

where we used that $\boldsymbol{g}_\mathrm{s}, \boldsymbol{g}_\mathrm{ns} \in \mathbb{R}^n$ have independent Gaussian entries. Here, $\boldsymbol{g} \in \mathbb{R}^n$ has i.i.d entries from $\mathsf{N}(0,1)$. Next, note that the above optimization over $\beta$ has a closed form. Using the identity $\max_{\beta \geq 0}(\beta x - \beta^2/2) = x_+^2/2$, with $x_+ = \max(x, 0)$, we get

$$
\begin{aligned}
\mathcal{L}_{AO}(\boldsymbol{\theta}_\mathrm{s}, \boldsymbol{\theta}_\mathrm{ns}) = \frac{1}{2n}\Big( & \left\| \boldsymbol{\varepsilon} + \boldsymbol{\Lambda}^\mathsf{T} \boldsymbol{M}_\mathrm{s}^\mathsf{T}(\boldsymbol{\theta}_{0,\mathrm{s}} - \boldsymbol{\theta}_\mathrm{s}) + \boldsymbol{\Lambda}^\mathsf{T} \boldsymbol{M}_\mathrm{ns}^\mathsf{T}(\boldsymbol{\theta}_{0,\mathrm{ns}} - \boldsymbol{\theta}_\mathrm{ns}) + \sqrt{\|\boldsymbol{\theta}_{0,\mathrm{s}}\|_{\ell_2}^2 + \|\boldsymbol{\theta}_{0,\mathrm{ns}} - \boldsymbol{\theta}_\mathrm{ns}\|_{\ell_2}^2}\, \boldsymbol{g} \right\|_{\ell_2} \\
& + \boldsymbol{h}_\mathrm{s}^\mathsf{T} \boldsymbol{\theta}_{0,\mathrm{s}} + \boldsymbol{h}_\mathrm{ns}^\mathsf{T}(\boldsymbol{\theta}_{0,\mathrm{ns}} - \boldsymbol{\theta}_\mathrm{ns}) \Big)_+^2 + \lambda \|\boldsymbol{\theta}_\mathrm{s}\|_{\ell_2}^2 + \lambda \|\boldsymbol{\theta}_\mathrm{ns}\|_{\ell_2}^2 \,.
\end{aligned} \tag{B.1}
$$

**Scalarization of the auxiliary optimization (AO) problem.** We next proceed to scalarize the AO problem. Consider the singular value decomposition

$$
\boldsymbol{M}_\mathrm{s} = \boldsymbol{U}_\mathrm{s} \boldsymbol{\Sigma}_\mathrm{s} \boldsymbol{V}_\mathrm{s}^\mathsf{T} \,,
$$

with $\boldsymbol{U}_\mathrm{s} \in \mathbb{R}^{p \times r}$, $\boldsymbol{\Sigma}_\mathrm{s} \in \mathbb{R}^{r \times r}$, $\boldsymbol{V}_\mathrm{s} \in \mathbb{R}^{k \times r}$, where $r = \mathrm{rank}(\boldsymbol{M}_\mathrm{s}) \leq k$. Decompose $\boldsymbol{q}_\mathrm{s} := \boldsymbol{\theta}_{0,\mathrm{s}} - \boldsymbol{\theta}_\mathrm{s}$ in its projections onto the space spanned by the columns $\boldsymbol{u}_{1,\mathrm{s}}, \ldots, \boldsymbol{u}_{r,\mathrm{s}}$ of $\boldsymbol{U}_\mathrm{s}$, and the orthogonal component:

$$
\boldsymbol{q}_\mathrm{s} = \sum_{i=1}^{r} \alpha_i \boldsymbol{u}_{i,\mathrm{s}} + \alpha_0 \boldsymbol{q}_\mathrm{s}^\perp \,,
$$

where $\|\boldsymbol{q}_\mathrm{s}^\perp\|_{\ell_2} = 1$, $\alpha_0 \geq 0$, and $\boldsymbol{U}_\mathrm{s}^\mathsf{T} \boldsymbol{q}_\mathrm{s}^\perp = \boldsymbol{0}$. Using the shorthand $\boldsymbol{\alpha} = (\alpha_1, \ldots, \alpha_r)$, we write

$$
\boldsymbol{\Lambda}^\mathsf{T} \boldsymbol{M}_\mathrm{s}^\mathsf{T}(\boldsymbol{\theta}_{0,\mathrm{s}} - \boldsymbol{\theta}_\mathrm{s}) = \boldsymbol{\Lambda}^\mathsf{T} \boldsymbol{V}_\mathrm{s} \boldsymbol{\Sigma}_\mathrm{s} \boldsymbol{U}_\mathrm{s}^\mathsf{T} \boldsymbol{q}_\mathrm{s} = \boldsymbol{\Lambda}^\mathsf{T} \boldsymbol{V}_\mathrm{s} \boldsymbol{\Sigma}_\mathrm{s} \boldsymbol{\alpha} \,.
$$

In addition,

$$
\begin{aligned}
\|\boldsymbol{\theta}_\mathrm{s}\|_{\ell_2}^2 &= \|\boldsymbol{\theta}_{0,\mathrm{s}} - (\boldsymbol{\theta}_{0,\mathrm{s}} - \boldsymbol{\theta}_\mathrm{s})\|_{\ell_2}^2 \\
&= \|\boldsymbol{\theta}_{0,\mathrm{s}}\|_{\ell_2}^2 + \|\boldsymbol{q}_\mathrm{s}\|_{\ell_2}^2 - 2\langle \boldsymbol{\theta}_{0,\mathrm{s}}, \boldsymbol{q}_\mathrm{s} \rangle \\
&= \|\boldsymbol{\theta}_{0,\mathrm{s}}\|_{\ell_2}^2 + \|\boldsymbol{q}_\mathrm{s}\|_{\ell_2}^2 - 2\langle \boldsymbol{\theta}_{0,\mathrm{s}}, \boldsymbol{U}_\mathrm{s} \boldsymbol{\alpha} \rangle - 2\alpha_0 \langle \boldsymbol{\theta}_{0,\mathrm{s}}, \boldsymbol{q}_\mathrm{s}^\perp \rangle \\
&= \|\boldsymbol{\theta}_{0,\mathrm{s}}\|_{\ell_2}^2 + \|\boldsymbol{q}_\mathrm{s}\|_{\ell_2}^2 - 2\langle \boldsymbol{U}_\mathrm{s}^\mathsf{T} \boldsymbol{\theta}_{0,\mathrm{s}}, \boldsymbol{\alpha} \rangle - 2\alpha_0 \langle \boldsymbol{\theta}_{0,\mathrm{s}}, \boldsymbol{q}_\mathrm{s}^\perp \rangle \\
&= \|\boldsymbol{\theta}_{0,\mathrm{s}}\|_{\ell_2}^2 + (\alpha_0^2 + \|\boldsymbol{\alpha}\|_{\ell_2}^2) - 2\langle \boldsymbol{U}_\mathrm{s}^\mathsf{T} \boldsymbol{\theta}_{0,\mathrm{s}}, \boldsymbol{\alpha} \rangle - 2\alpha_0 \langle \boldsymbol{\theta}_{0,\mathrm{s}}, \boldsymbol{q}_\mathrm{s}^\perp \rangle \,.
\end{aligned} \tag{B.2}
$$

Similarly, we define $\boldsymbol{q}_\mathrm{ns} = \boldsymbol{\theta}_{0,\mathrm{ns}} - \boldsymbol{\theta}_\mathrm{ns}$ and consider the singular value decomposition

$$
\boldsymbol{M}_\mathrm{ns} = \boldsymbol{U}_\mathrm{ns} \boldsymbol{\Sigma}_\mathrm{ns} \boldsymbol{V}_\mathrm{ns}^\mathsf{T} \,,
$$

with $\boldsymbol{U}_\mathrm{ns} \in \mathbb{R}^{(d-p) \times t}$, $\boldsymbol{\Sigma}_\mathrm{ns} \in \mathbb{R}^{t \times t}$, $\boldsymbol{V}_\mathrm{ns} \in \mathbb{R}^{k \times t}$, where $t = \mathrm{rank}(\boldsymbol{M}_\mathrm{ns}) \leq k$. Decomposing $\boldsymbol{q}_\mathrm{ns}$ in its projections on the orthogonal columns $\boldsymbol{u}_{1,\mathrm{ns}}, \ldots, \boldsymbol{u}_{r,\mathrm{ns}}$ of $\boldsymbol{U}_\mathrm{ns}$, and the orthogonal component we write

$$
\boldsymbol{q}_\mathrm{ns} = \sum_{i=1}^{t} \gamma_i \boldsymbol{u}_{i,\mathrm{ns}} + \gamma_0 \boldsymbol{q}_\mathrm{ns}^\perp \,,
$$

with $\|\boldsymbol{q}_\mathrm{ns}^\perp\|_{\ell_2} = 1$, $\gamma_0 \geq 0$, and $\boldsymbol{U}_\mathrm{ns}^\mathsf{T} \boldsymbol{q}_\mathrm{ns}^\perp = \boldsymbol{0}$. Define $\boldsymbol{\gamma} = (\gamma_1, \ldots, \gamma_t)$. In this notation, we have

$$
\boldsymbol{\Lambda}^\mathsf{T} \boldsymbol{M}_\mathrm{ns}^\mathsf{T}(\boldsymbol{\theta}_{0,\mathrm{ns}} - \boldsymbol{\theta}_\mathrm{ns}) = \boldsymbol{\Lambda}^\mathsf{T} \boldsymbol{V}_\mathrm{ns} \boldsymbol{\Sigma}_\mathrm{ns} \boldsymbol{U}_\mathrm{ns}^\mathsf{T} \boldsymbol{q}_\mathrm{ns} = \boldsymbol{\Lambda}^\mathsf{T} \boldsymbol{V}_\mathrm{ns} \boldsymbol{\Sigma}_\mathrm{ns} \boldsymbol{\gamma} \,.
$$

Also, $\|\boldsymbol{\theta}_{0,\mathrm{ns}} - \boldsymbol{\theta}_{\mathrm{ns}}\|_{\ell_2} = \|\boldsymbol{q}_{\mathrm{ns}}\|_{\ell_2} = \sqrt{\gamma_0^2 + \|\boldsymbol{\gamma}\|_{\ell_2}^2}$. In addition,

$$\boldsymbol{h}_{\mathrm{ns}}^{\mathsf{T}}(\boldsymbol{\theta}_{0,\mathrm{ns}} - \boldsymbol{\theta}_{\mathrm{ns}}) = \boldsymbol{h}_{\mathrm{ns}}^{\mathsf{T}}\boldsymbol{q}_{\mathrm{ns}} = \sum_{i=1}^{t} \gamma_i \boldsymbol{h}_{\mathrm{ns}}^{\mathsf{T}}\boldsymbol{u}_{i,\mathrm{ns}} + \gamma_0 \boldsymbol{h}_{\mathrm{ns}}^{\mathsf{T}}\boldsymbol{q}_{\mathrm{ns}}^{\perp} \,.$$

Using the above identities in (B.1), we have

$$\begin{aligned}
\mathcal{L}_{AO}(\boldsymbol{\theta}_{\mathrm{s}}, \boldsymbol{\theta}_{\mathrm{ns}}) = \frac{1}{2n} \Big( & \Big\| \boldsymbol{\varepsilon} + \boldsymbol{\Lambda}^{\mathsf{T}}\boldsymbol{V}_{\mathrm{s}}\boldsymbol{\Sigma}_{\mathrm{s}}\boldsymbol{\alpha} + \boldsymbol{\Lambda}^{\mathsf{T}}\boldsymbol{V}_{\mathrm{ns}}\boldsymbol{\Sigma}_{\mathrm{ns}}\boldsymbol{\gamma} + \sqrt{\|\boldsymbol{\theta}_{0,\mathrm{s}}\|_{\ell_2}^2 + \gamma_0^2 + \|\boldsymbol{\gamma}\|_{\ell_2}^2} \, \boldsymbol{g} \Big\|_{\ell_2} \\
& + \boldsymbol{h}_{\mathrm{s}}^{\mathsf{T}}\boldsymbol{\theta}_{0,\mathrm{s}} + \sum_{i=1}^{t} \gamma_i \boldsymbol{h}_{\mathrm{ns}}^{\mathsf{T}}\boldsymbol{u}_{i,\mathrm{ns}} + \gamma_0 \boldsymbol{h}_{\mathrm{ns}}^{\mathsf{T}}\boldsymbol{q}_{\mathrm{ns}}^{\perp} \Big)_{+}^{2} \\
& + \lambda \|\boldsymbol{\theta}_{0,\mathrm{s}}\|_{\ell_2}^2 + \lambda(\alpha_0^2 + \|\boldsymbol{\alpha}\|_{\ell_2}^2) - 2\lambda\langle \boldsymbol{U}_{\mathrm{s}}^{\mathsf{T}}\boldsymbol{\theta}_{0,\mathrm{s}}, \boldsymbol{\alpha}\rangle - 2\lambda\alpha_0\langle \boldsymbol{\theta}_{0,\mathrm{s}}, \boldsymbol{q}_{\mathrm{s}}^{\perp}\rangle \\
& + \lambda \|\boldsymbol{\theta}_{0,\mathrm{ns}}\|_{\ell_2}^2 + \lambda(\gamma_0^2 + \|\boldsymbol{\gamma}\|_{\ell_2}^2) - 2\lambda\langle \boldsymbol{U}_{\mathrm{ns}}^{\mathsf{T}}\boldsymbol{\theta}_{0,\mathrm{ns}}, \boldsymbol{\gamma}\rangle - 2\lambda\gamma_0\langle \boldsymbol{\theta}_{0,\mathrm{ns}}, \boldsymbol{q}_{\mathrm{ns}}^{\perp}\rangle . \quad \text{(B.3)}
\end{aligned}$$

By the above characterization, minimization over $\boldsymbol{\theta}_{\mathrm{s}}$ and $\boldsymbol{\theta}_{\mathrm{ns}}$ reduces to minimization over $\alpha_0, \gamma_0, \boldsymbol{\alpha}$, $\boldsymbol{\gamma}, \boldsymbol{q}_{\mathrm{s}}^{\perp}$ and $\boldsymbol{q}_{\mathrm{ns}}^{\perp}$. Further, these variables are free from each other and can be optimized over separately. For $\boldsymbol{q}_{\mathrm{s}}^{\perp}$, there is only one term involving this variable and therefore, minimization over it reduces to

$$\min_{\boldsymbol{q}_{\mathrm{s}}^{\perp}, \|\boldsymbol{q}_{\mathrm{s}}^{\perp}\|_{\ell_2}=1} -\langle \boldsymbol{\theta}_{0,\mathrm{s}}, \boldsymbol{q}_{\mathrm{s}}^{\perp}\rangle = \min_{\boldsymbol{q}_{\mathrm{s}}^{\perp}, \|\boldsymbol{q}_{\mathrm{s}}^{\perp}\|_{\ell_2}=1} -\langle \boldsymbol{U}_{\mathrm{s}}^{\perp}(\boldsymbol{U}_{\mathrm{s}}^{\perp})^{\mathsf{T}}\boldsymbol{\theta}_{0,\mathrm{s}}, \boldsymbol{q}_{\mathrm{s}}^{\perp}\rangle = -\big\|(\boldsymbol{U}_{\mathrm{s}}^{\perp})^{\mathsf{T}}\boldsymbol{\theta}_{0,\mathrm{s}}\big\|_{\ell_2} \,.$$

For $\boldsymbol{q}_{\mathrm{ns}}^{\perp}$, we note that there are two terms involving this variable, namely $\langle \frac{\boldsymbol{h}_{\mathrm{ns}}}{\sqrt{n}}, \boldsymbol{q}_{\mathrm{ns}}^{\perp}\rangle$ and $\langle (\boldsymbol{U}_{\mathrm{ns}}^{\perp})^{\mathsf{T}}\boldsymbol{\theta}_{0,\mathrm{ns}}, \boldsymbol{q}_{\mathrm{ns}}^{\perp}\rangle$. Since $\|\boldsymbol{q}_{\mathrm{ns}}^{\perp}\|_{\ell_2} = 1$, it is easy to see that the optimal $\boldsymbol{q}_{\mathrm{ns}}^{\perp}$ should be in the span of $\boldsymbol{h}_{\mathrm{ns}}^{\perp}$ and $(\boldsymbol{U}_{\mathrm{ns}}^{\perp})^{\mathsf{T}}\boldsymbol{\theta}_{0,\mathrm{ns}}$. In addition,

$$\langle \frac{\boldsymbol{h}_{\mathrm{ns}}^{\perp}}{\sqrt{n}}, (\boldsymbol{U}_{\mathrm{ns}}^{\perp})^{\mathsf{T}}\boldsymbol{\theta}_{0,\mathrm{ns}}\rangle \overset{(p)}{\to} 0 \,,$$

by the law of large numbers. In words, these two vectors are asymptotically orthogonal. Hence, we can consider the following decomposition of the optimal $\boldsymbol{q}_{\mathrm{ns}}^{\perp}$:

$$\boldsymbol{q}_{\mathrm{ns}}^{\perp} = -\xi \frac{\boldsymbol{h}_{\mathrm{ns}}^{\perp}}{\|\boldsymbol{h}_{\mathrm{ns}}^{\perp}\|_{\ell_2}} + \sqrt{1 - \xi^2} \frac{\boldsymbol{U}_{\mathrm{ns}}^{\perp}(\boldsymbol{U}_{\mathrm{ns}}^{\perp})^{\mathsf{T}}\boldsymbol{\theta}_{0,\mathrm{ns}}}{\|(\boldsymbol{U}_{\mathrm{ns}}^{\perp})^{\mathsf{T}}\boldsymbol{\theta}_{0,\mathrm{ns}}\|_{\ell_2}} \,,$$

where $\xi \geq 0$ and $\boldsymbol{h}_{\mathrm{ns}}^{\perp}$ denotes the projection of $\boldsymbol{h}_{\mathrm{ns}}$ onto the (left) null space of $\boldsymbol{U}_{\mathrm{ns}}$. This brings us to

$$\begin{aligned}
\min_{\boldsymbol{\theta}_{\mathrm{s}}, \boldsymbol{\theta}_{\mathrm{ns}}} \mathcal{L}_{AO}(\boldsymbol{\theta}_{\mathrm{s}}, \boldsymbol{\theta}_{\mathrm{ns}}) = \min_{\alpha_0, \gamma_0 \geq 0, \boldsymbol{\alpha}, \boldsymbol{\gamma}} \frac{1}{2} \Big( & \frac{1}{\sqrt{n}} \Big\| \boldsymbol{\varepsilon} + \boldsymbol{\Lambda}^{\mathsf{T}}\boldsymbol{V}_{\mathrm{s}}\boldsymbol{\Sigma}_{\mathrm{s}}\boldsymbol{\alpha} + \boldsymbol{\Lambda}^{\mathsf{T}}\boldsymbol{V}_{\mathrm{ns}}\boldsymbol{\Sigma}_{\mathrm{ns}}\boldsymbol{\gamma} + \sqrt{\|\boldsymbol{\theta}_{0,\mathrm{s}}\|_{\ell_2}^2 + \gamma_0^2 + \|\boldsymbol{\gamma}\|_{\ell_2}^2} \, \boldsymbol{g} \Big\|_{\ell_2} \\
& + \frac{\boldsymbol{h}_{\mathrm{s}}^{\mathsf{T}}\boldsymbol{\theta}_{0,\mathrm{s}}}{\sqrt{n}} + \sum_{i=1}^{t} \gamma_i \frac{\boldsymbol{h}_{\mathrm{ns}}^{\mathsf{T}}\boldsymbol{u}_{i,\mathrm{ns}}}{\sqrt{n}} - \gamma_0 \xi \frac{\|\boldsymbol{h}_{\mathrm{ns}}^{\perp}\|_{\ell_2}}{\sqrt{n}} \Big)_{+}^{2} \\
& + \lambda \|\boldsymbol{\theta}_{0,\mathrm{s}}\|_{\ell_2}^2 + \lambda(\alpha_0^2 + \|\boldsymbol{\alpha}\|_{\ell_2}^2) - 2\lambda\langle \boldsymbol{U}_{\mathrm{s}}^{\mathsf{T}}\boldsymbol{\theta}_{0,\mathrm{s}}, \boldsymbol{\alpha}\rangle - 2\lambda\alpha_0 \big\|(\boldsymbol{U}_{\mathrm{s}}^{\perp})^{\mathsf{T}}\boldsymbol{\theta}_{0,\mathrm{s}}\big\|_{\ell_2} \\
& + \lambda \|\boldsymbol{\theta}_{0,\mathrm{ns}}\|_{\ell_2}^2 + \lambda(\gamma_0^2 + \|\boldsymbol{\gamma}\|_{\ell_2}^2) - 2\lambda\langle \boldsymbol{U}_{\mathrm{ns}}^{\mathsf{T}}\boldsymbol{\theta}_{0,\mathrm{ns}}, \boldsymbol{\gamma}\rangle - 2\lambda\gamma_0\sqrt{1 - \xi^2} \big\|(\boldsymbol{U}_{\mathrm{ns}}^{\perp})^{\mathsf{T}}\boldsymbol{\theta}_{0,\mathrm{ns}}\big\|_{\ell_2} \,.
\end{aligned}$$
$$\text{(B.4)}$$

Note that at this stage, the AO problem is reduced to an optimization over $r + t + 3$ scalar variables $(\alpha_0, \gamma_0 \geq 0, 0 \leq \xi \leq 1$ and $\boldsymbol{\alpha} \in \mathbb{R}^r, \boldsymbol{\gamma} \in \mathbb{R}^t)$.

**Convergence of the auxiliary optimization problem.** We next continue to derive the point-wise in-probability limit of the AO problem.

First observe that since $\boldsymbol{\varepsilon}$ and $\boldsymbol{g}$ are independent with i.i.d $\mathsf{N}(0,1)$ entries, we have

$$\boldsymbol{\varepsilon} + \sqrt{\|\boldsymbol{\theta}_{0,\mathrm{s}}\|_{\ell_2}^2 + \gamma_0^2 + \|\boldsymbol{\gamma}\|_{\ell_2}^2} \, \boldsymbol{g} \overset{(d)}{=} \sqrt{\sigma^2 + \,^2 (\|\boldsymbol{\theta}_{0,\mathrm{s}}\|_{\ell_2}^2 + \gamma_0^2 + \|\boldsymbol{\gamma}\|_{\ell_2}^2)} \, \tilde{\boldsymbol{g}} \,,$$

where $\tilde{\boldsymbol{g}} \in \mathbb{R}^n$ has i.i.d $\mathsf{N}(0,1)$ entries.

Second, by construction $\boldsymbol{\Lambda}\boldsymbol{\Lambda}^{\mathsf{T}} = \mathrm{diag}(n_1, \ldots, n_k) \in \mathbb{R}^{k \times k}$, where $n_\ell$ denotes the number of examples from cluster $\ell$. Hence,

$$\begin{aligned}
\frac{1}{n} \big\| \boldsymbol{\Lambda}^{\mathsf{T}}\boldsymbol{V}_{\mathrm{s}}\boldsymbol{\Sigma}_{\mathrm{s}}\boldsymbol{\alpha} + \boldsymbol{\Lambda}^{\mathsf{T}}\boldsymbol{V}_{\mathrm{ns}}\boldsymbol{\Sigma}_{\mathrm{ns}}\boldsymbol{\gamma} \big\|_{\ell_2}^2 &= (\boldsymbol{V}_{\mathrm{s}}\boldsymbol{\Sigma}_{\mathrm{s}}\boldsymbol{\alpha} + \boldsymbol{V}_{\mathrm{ns}}\boldsymbol{\Sigma}_{\mathrm{ns}}\boldsymbol{\gamma})^{\mathsf{T}} \mathrm{diag}(\tfrac{n_1}{n}, \ldots, \tfrac{n_k}{k})(\boldsymbol{V}_{\mathrm{s}}\boldsymbol{\Sigma}_{\mathrm{s}}\boldsymbol{\alpha} + \boldsymbol{V}_{\mathrm{ns}}\boldsymbol{\Sigma}_{\mathrm{ns}}\boldsymbol{\gamma}) \\
&\overset{(p)}{\to} (\boldsymbol{V}_{\mathrm{s}}\boldsymbol{\Sigma}_{\mathrm{s}}\boldsymbol{\alpha} + \boldsymbol{V}_{\mathrm{ns}}\boldsymbol{\Sigma}_{\mathrm{ns}}\boldsymbol{\gamma})^{\mathsf{T}} \mathrm{diag}(\boldsymbol{\pi})(\boldsymbol{V}_{\mathrm{s}}\boldsymbol{\Sigma}_{\mathrm{s}}\boldsymbol{\alpha} + \boldsymbol{V}_{\mathrm{ns}}\boldsymbol{\Sigma}_{\mathrm{ns}}\boldsymbol{\gamma})
\end{aligned}$$

Next, by using concentration of Lipschitz functions of Gaussian vectors, we obtain

$$\frac{1}{\sqrt{n}} \left\| \boldsymbol{\varepsilon} + \boldsymbol{\Lambda}^\mathsf{T} \boldsymbol{V}_\mathrm{s} \boldsymbol{\Sigma}_\mathrm{s} \boldsymbol{\alpha} + \boldsymbol{\Lambda}^\mathsf{T} \boldsymbol{V}_\mathrm{ns} \boldsymbol{\Sigma}_\mathrm{ns} \boldsymbol{\gamma} + \sqrt{\|\boldsymbol{\theta}_{0,\mathrm{s}}\|_{\ell_2}^2 + \gamma_0^2 + \|\boldsymbol{\gamma}\|_{\ell_2}^2} \, \boldsymbol{g} \right\|_{\ell_2}$$

$$\overset{p}{\to} \sqrt{(\boldsymbol{V}_\mathrm{s} \boldsymbol{\Sigma}_\mathrm{s} \boldsymbol{\alpha} + \boldsymbol{V}_\mathrm{ns} \boldsymbol{\Sigma}_\mathrm{ns} \boldsymbol{\gamma})^\mathsf{T} \mathrm{diag}(\boldsymbol{\pi})(\boldsymbol{V}_\mathrm{s} \boldsymbol{\Sigma}_\mathrm{s} \boldsymbol{\alpha} + \boldsymbol{V}_\mathrm{ns} \boldsymbol{\Sigma}_\mathrm{ns} \boldsymbol{\gamma}) + \sigma^2 + (\|\boldsymbol{\theta}_{0,\mathrm{s}}\|_{\ell_2}^2 + \gamma_0^2 + \|\boldsymbol{\gamma}\|_{\ell_2}^2)}$$

Also, since $\|\boldsymbol{\theta}_{0,\mathrm{s}}\|_{\ell_2}$ is bounded and $\|\boldsymbol{u}_{i,\mathrm{s}}\|_{\ell_2} = 1$, we get

$$\frac{\boldsymbol{h}_\mathrm{s}^\mathsf{T} \boldsymbol{\theta}_{0,\mathrm{s}}}{\sqrt{n}}, \frac{\boldsymbol{h}_\mathrm{ns}^\mathsf{T} \boldsymbol{u}_{i,\mathrm{ns}}}{\sqrt{n}} \overset{(p)}{\to} 0 \,.$$

In addition, $\|\boldsymbol{h}_\mathrm{ns}^\perp\|_{\ell_2}$ concentrates around $\sqrt{d-p-t}$ and $(d-p-t)/n \to \psi_d - \psi_p$, because $t \leq k$ remains bounded as $n$ diverges, and so

$$\frac{\|\boldsymbol{h}_\mathrm{ns}^\perp\|_{\ell_2}}{\sqrt{n}} \overset{(p)}{\to} \sqrt{\psi_d - \psi_p} \,.$$

Using the above limits, the objective in (B.4) converges in-probability to

$$\mathcal{D}(\alpha_0, \gamma_0, \xi, \boldsymbol{\alpha}, \boldsymbol{\gamma}) :=$$

$$\frac{1}{2} \left( \sqrt{(\boldsymbol{V}_\mathrm{s} \boldsymbol{\Sigma}_\mathrm{s} \boldsymbol{\alpha} + \boldsymbol{V}_\mathrm{ns} \boldsymbol{\Sigma}_\mathrm{ns} \boldsymbol{\gamma})^\mathsf{T} \mathrm{diag}(\boldsymbol{\pi})(\boldsymbol{V}_\mathrm{s} \boldsymbol{\Sigma}_\mathrm{s} \boldsymbol{\alpha} + \boldsymbol{V}_\mathrm{ns} \boldsymbol{\Sigma}_\mathrm{ns} \boldsymbol{\gamma}) + \sigma^2 + (\|\boldsymbol{\theta}_{0,\mathrm{s}}\|_{\ell_2}^2 + \gamma_0^2 + \|\boldsymbol{\gamma}\|_{\ell_2}^2)} - \gamma_0 \xi \sqrt{\psi_d - \psi_p} \right)_+^2$$

$$+ \lambda \|\boldsymbol{\theta}_0\|_{\ell_2}^2 + \lambda(\alpha_0^2 + \gamma_0^2 + \|\boldsymbol{\alpha}\|_{\ell_2}^2 + \|\boldsymbol{\gamma}\|_{\ell_2}^2)$$

$$- 2\lambda \left( \langle \boldsymbol{U}_\mathrm{s}^\mathsf{T} \boldsymbol{\theta}_{0,\mathrm{s}}, \boldsymbol{\alpha} \rangle + \alpha_0 \|(\boldsymbol{U}_\mathrm{s}^\perp)^\mathsf{T} \boldsymbol{\theta}_{0,\mathrm{s}}\|_{\ell_2} + \langle \boldsymbol{U}_\mathrm{ns}^\mathsf{T} \boldsymbol{\theta}_{0,\mathrm{ns}}, \boldsymbol{\gamma} \rangle + \gamma_0 \sqrt{1 - \xi^2} \|(\boldsymbol{U}_\mathrm{ns}^\perp)^\mathsf{T} \boldsymbol{\theta}_{0,\mathrm{ns}}\|_{\ell_2} \right) \tag{B.5}$$

We are now ready to prove the theorems.

### B.2.1 Proof of Theorem 3.1

Using Lemma 2.1, we have

$$\begin{aligned}
\mathrm{Risk}(\widehat{\boldsymbol{\theta}}_L) &= \sigma^2 + \|\boldsymbol{\theta}_0 - \boldsymbol{\theta}\|_{\ell_2}^2 + (\boldsymbol{\theta}_0 - \boldsymbol{\theta})^\mathsf{T} \boldsymbol{M} \mathrm{diag}(\boldsymbol{\pi}) \boldsymbol{M}^\mathsf{T} (\boldsymbol{\theta}_0 - \boldsymbol{\theta}) \\
&= \sigma^2 + \|\boldsymbol{q}_\mathrm{s}\|_{\ell_2}^2 + \|\boldsymbol{q}_\mathrm{ns}\|_{\ell_2}^2 + \boldsymbol{q}_\mathrm{s}^\mathsf{T} \boldsymbol{M}_\mathrm{s} \mathrm{diag}(\boldsymbol{\pi}) \boldsymbol{M}_\mathrm{s}^\mathsf{T} \boldsymbol{q}_\mathrm{s} + \boldsymbol{q}_\mathrm{ns}^\mathsf{T} \boldsymbol{M}_\mathrm{ns} \mathrm{diag}(\boldsymbol{\pi}) \boldsymbol{M}_\mathrm{ns}^\mathsf{T} \boldsymbol{q}_\mathrm{ns} \\
&= \sigma^2 + (\alpha_0^2 + \gamma_0^2 + \|\boldsymbol{\alpha}\|_{\ell_2}^2 + \|\boldsymbol{\gamma}\|_{\ell_2}^2) \\
&\quad + \boldsymbol{\alpha}^\mathsf{T} \boldsymbol{\Sigma}_\mathrm{s} \boldsymbol{V}_\mathrm{s}^\mathsf{T} \mathrm{diag}(\boldsymbol{\pi}) \boldsymbol{V}_\mathrm{s} \boldsymbol{\Sigma}_\mathrm{s} \boldsymbol{\alpha} + \boldsymbol{\gamma}^\mathsf{T} \boldsymbol{\Sigma}_\mathrm{ns} \boldsymbol{V}_\mathrm{ns}^\mathsf{T} \mathrm{diag}(\boldsymbol{\pi}) \boldsymbol{V}_\mathrm{ns} \boldsymbol{\Sigma}_\mathrm{ns} \boldsymbol{\gamma} \,. \tag{B.6}
\end{aligned}$$

Since $\psi_d - \psi_p \leq 1$, we are in the over- determined (a.k.a underparametrized) regime. As $\lambda \to 0^+$, the terms involving $\lambda$ become negligible compared to the first term in (B.5) except those that include $\alpha_0$, as $\alpha_0$ is not present in the first term . Since $(x)_+^2$ is increasing, and

$$(\boldsymbol{V}_\mathrm{s} \boldsymbol{\Sigma}_\mathrm{s} \boldsymbol{\alpha} + \boldsymbol{V}_\mathrm{ns} \boldsymbol{\Sigma}_\mathrm{ns} \boldsymbol{\gamma})^\mathsf{T} \mathrm{diag}(\boldsymbol{\pi})(\boldsymbol{V}_\mathrm{s} \boldsymbol{\Sigma}_\mathrm{s} \boldsymbol{\alpha} + \boldsymbol{V}_\mathrm{ns} \boldsymbol{\Sigma}_\mathrm{ns} \boldsymbol{\gamma}) + \|\boldsymbol{\gamma}\|_{\ell_2}^2 \geq 0,$$

the minimum over $\boldsymbol{\alpha}$ and $\boldsymbol{\gamma}$ is achieved for $\boldsymbol{\alpha} = \boldsymbol{0} \in \mathbb{R}^r$ and $\boldsymbol{\gamma} = \boldsymbol{0} \in \mathbb{R}^t$. The optimization (B.5) then reduces to

$$\min_{\alpha_0, \gamma_0 \geq 0, 0 \leq \xi \leq 1} \frac{1}{2} \left( \sqrt{\sigma^2 + (\|\boldsymbol{\theta}_{0,\mathrm{s}}\|_{\ell_2}^2 + \gamma_0^2)} - \gamma_0 \xi \sqrt{\psi_d - \psi_p} \right)_+^2 + \lambda \alpha_0^2 - 2\lambda \alpha_0 \|(\boldsymbol{U}_\mathrm{s}^\perp)^\mathsf{T} \boldsymbol{\theta}_{0,\mathrm{s}}\|_{\ell_2} \,. \tag{B.7}$$

The optimal $\xi$ is given by $\xi = 1$. Also, setting derivative with respect to $\alpha_0$ to zero we obtain the optimal $\alpha_0 = \|(\boldsymbol{U}_\mathrm{s}^\perp)^\mathsf{T} \boldsymbol{\theta}_{0,\mathrm{s}}\|_{\ell_2}$. Next, by setting derivative with respect to $\gamma_0$ we arrive at

$$\gamma_0^2 = (\sigma^2 + \|\boldsymbol{\theta}_{0,\mathrm{s}}\|_{\ell_2}^2) \frac{\psi_d - \psi_p}{1 - (\psi_d - \psi_p)} \,.$$

Using the optimal variables in (B.6) we obtain the risk of minimum-norm estimator as

$$\begin{aligned}
\mathrm{Risk}(\widehat{\boldsymbol{\theta}}_L) &= \sigma^2 + \|(\boldsymbol{U}_\mathrm{s}^\perp)^\mathsf{T} \boldsymbol{\theta}_{0,\mathrm{s}}\|_{\ell_2}^2 + (\sigma^2 + \|\boldsymbol{\theta}_{0,\mathrm{s}}\|_{\ell_2}^2) \frac{\psi_d - \psi_p}{1 - (\psi_d - \psi_p)} \\
&= (\sigma^2 + \|\boldsymbol{\theta}_{0,\mathrm{s}}\|_{\ell_2}^2) \frac{1}{1 - (\psi_d - \psi_p)} - \|\boldsymbol{U}_\mathrm{s}^\mathsf{T} \boldsymbol{\theta}_{0,\mathrm{s}}\|_{\ell_2}^2 \,.
\end{aligned}$$

Recall that by assumption, $r_\mathrm{s} = \|\boldsymbol{\theta}_{0,\mathrm{s}}\|_{\ell_2}$ and $\|\boldsymbol{U}_\mathrm{s}^\mathsf{T} \boldsymbol{\theta}_{0,\mathrm{s}}\|_{\ell_2} = \sqrt{\rho} r_\mathrm{s}$, which completes the proof.

### B.2.2 Proof of Theorem 3.2

We continue from (B.5). In the case of $\psi_d - \psi_p \leq 1$, it is easy to see that the derivative of the first term of (B.5), in the active region is decreasing in $\gamma_0$. With the consideration $\lambda \to 0^+$, minimizing over $\gamma_0$ will push us into the non-active region. Therefore the optimization problem (B.5) reduces to

$$\text{minimize} \quad \|\boldsymbol{\theta}_0\|_{\ell_2}^2 + \alpha_0^2 + \gamma_0^2 + \|\boldsymbol{\alpha}\|_{\ell_2}^2 + \|\boldsymbol{\gamma}\|_{\ell_2}^2$$
$$- 2\left(\langle \boldsymbol{U}_s^\top \boldsymbol{\theta}_{0,s}, \boldsymbol{\alpha}\rangle + \alpha_0 \left\|(\boldsymbol{U}_s^\perp)^\top \boldsymbol{\theta}_{0,s}\right\|_{\ell_2} + \langle \boldsymbol{U}_{ns}^\top \boldsymbol{\theta}_{0,ns}, \boldsymbol{\gamma}\rangle + \gamma_0\sqrt{1-\xi^2}\left\|(\boldsymbol{U}_{ns}^\perp)^\top \boldsymbol{\theta}_{0,ns}\right\|_{\ell_2}\right)$$

subject to

$$(\boldsymbol{V}_s\boldsymbol{\Sigma}_s\boldsymbol{\alpha} + \boldsymbol{V}_{ns}\boldsymbol{\Sigma}_{ns}\boldsymbol{\gamma})^\top \text{diag}(\boldsymbol{\pi})(\boldsymbol{V}_s\boldsymbol{\Sigma}_s\boldsymbol{\alpha} + \boldsymbol{V}_{ns}\boldsymbol{\Sigma}_{ns}\boldsymbol{\gamma}) + \sigma^2 + (\|\boldsymbol{\theta}_{0,s}\|_{\ell_2}^2 + \gamma_0^2 + \|\boldsymbol{\gamma}\|_{\ell_2}^2) \leq \gamma_0^2\xi^2(\psi_d - \psi_p)$$
$$\text{(B.8)}$$

By Assumption 2, $\boldsymbol{\Sigma}_s = \mu\boldsymbol{I}_k$, $\boldsymbol{V}_s = \boldsymbol{I}_{k\times k}$, and $\boldsymbol{\Sigma}_{ns} = \boldsymbol{0}$, $\boldsymbol{U}_{ns} = \boldsymbol{0}$ (no cluster structure on non-sensitive features and an orthogonal, equal energy cluster centers on the sensitive features). Therefore, by fixing $\gamma := \|\boldsymbol{\gamma}\|_{\ell_2}$, the optimization problem (B.8) becomes:

$$\text{minimize} \quad \alpha_0^2 + \gamma_0^2 + \|\boldsymbol{\alpha}\|_{\ell_2}^2 + \gamma^2 - 2\left(\langle \boldsymbol{U}_s^\top\boldsymbol{\theta}_{0,s}, \boldsymbol{\alpha}\rangle + \alpha_0\left\|(\boldsymbol{U}_s^\perp)^\top\boldsymbol{\theta}_{0,s}\right\|_{\ell_2} + \gamma_0\sqrt{1-\xi^2}\left\|\boldsymbol{\theta}_{0,ns}\right\|_{\ell_2}\right)$$

subject to

$$\mu^2\boldsymbol{\alpha}^\top\text{diag}(\boldsymbol{\pi})\boldsymbol{\alpha} + \sigma^2 + \|\boldsymbol{\theta}_{0,s}\|_{\ell_2}^2 + \gamma_0^2 + \gamma^2 \leq \gamma_0^2\xi^2(\psi_d - \psi_p). \tag{B.9}$$

Since $\alpha_0$ does not appear in the constraint, it is easy to see that its optimal value is given by $\alpha_0 = \left\|(\boldsymbol{U}_s^\perp)^\top\boldsymbol{\theta}_{0,s}\right\|_{\ell_2}$. Also, note that by decreasing $\gamma$ the objective value decreases and also by the constraint on the other variables become more relaxed. Consequently, the optimal value of $\gamma$ is $\gamma = 0$. Removing $\alpha_0$ from the objective function, we are left with

$$\text{minimize} \quad \gamma_0^2 + \|\boldsymbol{\alpha}\|_{\ell_2}^2 - 2\left(\langle \boldsymbol{U}_s^\top\boldsymbol{\theta}_{0,s}, \boldsymbol{\alpha}\rangle + \gamma_0\sqrt{1-\xi^2}\left\|\boldsymbol{\theta}_{0,ns}\right\|_{\ell_2}\right)$$
$$\text{subject to} \quad \mu^2\boldsymbol{\alpha}^\top\text{diag}(\boldsymbol{\pi})\boldsymbol{\alpha} + \sigma^2 + \|\boldsymbol{\theta}_{0,s}\|_{\ell_2}^2 + \gamma_0^2 \leq \gamma_0^2\xi^2(\psi_d - \psi_p). \tag{B.10}$$

Optimal choice of $\xi$ results in the constraint to become equality. Solving for $\xi$, the optimization reduces to

$$\text{minimize} \quad \gamma_0^2 + \|\boldsymbol{\alpha}\|_{\ell_2}^2 - 2\left(\langle \boldsymbol{U}_s^\top\boldsymbol{\theta}_{0,s}, \boldsymbol{\alpha}\rangle + \sqrt{\gamma_0^2 - \frac{\mu^2\boldsymbol{\alpha}^\top\text{diag}(\boldsymbol{\pi})\boldsymbol{\alpha} + \sigma^2 + \|\boldsymbol{\theta}_{0,s}\|_{\ell_2}^2 + \gamma_0^2}{\psi_d - \psi_p}}\left\|\boldsymbol{\theta}_{0,ns}\right\|_{\ell_2}\right)$$

Setting derivative with respect to $\gamma_0$ to zero, we obtain

$$\sqrt{\gamma_0^2 - \frac{\mu^2\boldsymbol{\alpha}^\top\text{diag}(\boldsymbol{\pi})\boldsymbol{\alpha} + \sigma^2 + \|\boldsymbol{\theta}_{0,s}\|_{\ell_2}^2 + \gamma_0^2}{\psi_d - \psi_p}} = \left(1 - \frac{1}{\psi_d - \psi_p}\right)\left\|\boldsymbol{\theta}_{0,ns}\right\|_{\ell_2}. \tag{B.11}$$

Setting derivative with respect to $\boldsymbol{\alpha}$ to zero and using the previous stationary equation, we get

$$\boldsymbol{\alpha} = \left(\boldsymbol{I} + \frac{\mu^2\text{diag}(\boldsymbol{\pi})}{\psi_d - \psi_p - 1}\right)^{-1}\boldsymbol{U}_s^\top\boldsymbol{\theta}_{0,s}. \tag{B.12}$$

We next square both sides of (B.12) and rearrange the terms to get

$$\gamma_0^2 = \frac{1}{\psi_d - \psi_p - 1}\left(\sigma^2 + \|\boldsymbol{\theta}_{0,s}\|_{\ell_2}^2 + \mu^2\boldsymbol{\alpha}^\top\text{diag}(\boldsymbol{\pi})\boldsymbol{\alpha}\right) + \left(1 - \frac{1}{\psi_d - \psi_p}\right)\left\|\boldsymbol{\theta}_{0,ns}\right\|_{\ell_2}^2$$

$$= \frac{1}{\psi_d - \psi_p - 1}\left(\sigma^2 + r_s^2 + \mu^2\boldsymbol{\alpha}^\top\text{diag}(\boldsymbol{\pi})\boldsymbol{\alpha}\right) + \left(1 - \frac{1}{\psi_d - \psi_p}\right)r_{ns}^2,$$

which are the same expressions for $\boldsymbol{\alpha}$ and $\gamma_0$ given in the theorem statement.

The final step is to write the risk of estimator in terms of $\boldsymbol{\alpha}$, $\gamma_0$. Invoke equation (B.6), and recall that in the current case, $\boldsymbol{\Sigma}_{ns} = \boldsymbol{0}$, $\boldsymbol{\Sigma}_s = \mu\boldsymbol{I}$. Also, as we showed in our derivation, $\gamma = \|\boldsymbol{\gamma}\|_{\ell_2} = 0$, $\alpha_0 = \left\|(\boldsymbol{U}_s^\perp)^\top\boldsymbol{\theta}_{0,s}\right\|_{\ell_2}$, by which we arrive at

$$\text{Risk}(\widehat{\boldsymbol{\theta}}_L) = \mu^2\boldsymbol{\alpha}^\top\text{diag}(\boldsymbol{\pi})\boldsymbol{\alpha} + \sigma^2 + (\left\|(\boldsymbol{U}_s^\perp)^\top\boldsymbol{\theta}_{0,s}\right\|_{\ell_2}^2 + \gamma_0^2 + \|\boldsymbol{\alpha}\|_{\ell_2}^2)$$
$$= \sigma^2 + (1-\rho)r_s^2 + \gamma_0^2 + \boldsymbol{\alpha}^\top\left(\boldsymbol{I} + \mu^2\text{diag}(\boldsymbol{\pi})\right)\boldsymbol{\alpha}. \tag{B.13}$$

This concludes the proof.

## B.3 Proof of Theorem 3.3

We follow the proof strategy used for Theorem 3.1-3.2. Here, we would like to characterize the risk of min-norm estimator $\widehat{\boldsymbol{\theta}}$. The features matrix has a clustering structure, but the learner is not using that (no look-alike clustering) and is just compute the min-norm estimator for fitting the responses to individual features. Therefore, one can think of this setting as a special case of our previous analysis when there is no sensitive features (so $\psi_p = 0$).

$(a)$ By setting $\psi_p = 0$ and $r_{\text{s}} = 0$ in the result of Theorem 3.1, we get that when $\psi_d \le 1$,

$$\text{Risk}(\widehat{\boldsymbol{\theta}}) = \frac{\sigma^2}{1 - \psi_d}\,.$$

$(b)$ In this case, we specialize the proof of Theorem 3.2 to the case that $\psi_p = 0$. Continuing from (B.8), and removing the terms corresponding to sensitive features, we arrive at

$$\text{minimize} \quad \gamma_0^2 + \|\boldsymbol{\gamma}\|_{\ell_2}^2 - 2\left(\langle \boldsymbol{U}_{\text{ns}}^\mathsf{T}\boldsymbol{\theta}_{0,\text{ns}}, \boldsymbol{\gamma}\rangle + \gamma_0\sqrt{1-\xi^2}\left\|(\boldsymbol{U}_{\text{ns}}^\perp)^\mathsf{T}\boldsymbol{\theta}_{0,\text{ns}}\right\|_{\ell_2}\right)$$

subject to

$$(\boldsymbol{V}_{\text{ns}}\boldsymbol{\Sigma}_{\text{ns}}\boldsymbol{\gamma})^\mathsf{T}\text{diag}(\boldsymbol{\pi})(\boldsymbol{V}_{\text{ns}}\boldsymbol{\Sigma}_{\text{ns}}\boldsymbol{\gamma}) + \sigma^2 + \gamma_0^2 + \|\boldsymbol{\gamma}\|_{\ell_2}^2 \le \gamma_0^2\xi^2\psi_d \qquad \text{(B.14)}$$

We drop the index 'ns' as it is not relevant in this case. Also by Assumption 2, $\boldsymbol{\Sigma}_{\text{ns}} = \mu\boldsymbol{I}_d$, $\boldsymbol{V}_{\text{ns}} = \boldsymbol{I}_d$. Therefore, the above optimization can be written as

$$\text{minimize} \quad \gamma_0^2 + \|\boldsymbol{\gamma}\|_{\ell_2}^2 - 2\left(\langle \boldsymbol{U}^\mathsf{T}\boldsymbol{\theta}_0, \boldsymbol{\gamma}\rangle + \gamma_0\sqrt{1-\xi^2}\left\|(\boldsymbol{U}^\perp)^\mathsf{T}\boldsymbol{\theta}_0\right\|_{\ell_2}\right)$$

$$\text{subject to} \quad \boldsymbol{\gamma}^\mathsf{T}(\boldsymbol{I} + \mu^2\text{diag}(\boldsymbol{\pi}))\boldsymbol{\gamma} + \sigma^2 + \gamma_0^2 \le \gamma_0^2\xi^2\psi_d\,. \qquad \text{(B.15)}$$

Optimal $\xi$ makes the constraint equality. Solving for $\xi$, the above optimization can be written as so we have

$$\text{minimize} \quad \gamma_0^2 + \|\boldsymbol{\gamma}\|_{\ell_2}^2 - 2\left(\langle \boldsymbol{U}^\mathsf{T}\boldsymbol{\theta}_0, \boldsymbol{\gamma}\rangle + \sqrt{\gamma_0^2 - \frac{\boldsymbol{\gamma}^\mathsf{T}(\boldsymbol{I} + \mu^2\text{diag}(\boldsymbol{\pi}))\boldsymbol{\gamma} + \sigma^2 + \gamma_0^2}{\psi_d}}\left\|(\boldsymbol{U}^\perp)^\mathsf{T}\boldsymbol{\theta}_0\right\|_{\ell_2}\right)\,.$$

Setting the derivative with respect to $\gamma_0$ to zero, we get

$$\sqrt{\gamma_0^2 - \frac{\boldsymbol{\gamma}^\mathsf{T}(\boldsymbol{I} + \mu^2\text{diag}(\boldsymbol{\pi}))\boldsymbol{\gamma} + \sigma^2 + \gamma_0^2}{\psi_d}} = \left(1 - \frac{1}{\psi_d}\right)\left\|(\boldsymbol{U}^\perp)^\mathsf{T}\boldsymbol{\theta}_0\right\|_{\ell_2}\,. \qquad \text{(B.16)}$$

Setting derivative with respect to $\boldsymbol{\gamma}$ to zero and using the above equation, we obtain

$$\boldsymbol{\gamma} = \left(\boldsymbol{I} + \frac{\boldsymbol{I} + \mu^2\text{diag}(\boldsymbol{\pi})}{\psi_d - 1}\right)^{-1}\boldsymbol{U}^\mathsf{T}\boldsymbol{\theta}_0\,. \qquad \text{(B.17)}$$

We next square both sides of equation (B.16), and rearrange the terms to get:

$$\gamma_0^2 = \frac{1}{\psi_d - 1}\left(\sigma^2 + \boldsymbol{\gamma}^\mathsf{T}(\boldsymbol{I} + \mu^2\text{diag}(\boldsymbol{\pi}))\boldsymbol{\gamma}\right) + \left(1 - \frac{1}{\psi_d}\right)\left\|(\boldsymbol{U}^\perp)^\mathsf{T}\boldsymbol{\theta}_0\right\|_{\ell_2}^2\,.$$

Under the simplifying Assumption 2, there is no cluster structure on the non-sensitive features and so $\boldsymbol{U}_{\text{ns}} = 0$. Therefore,

$$\left\|\boldsymbol{U}^\mathsf{T}\boldsymbol{\theta}_0\right\|_{\ell_2} = \left\|\boldsymbol{U}_{\text{s}}^\mathsf{T}\boldsymbol{\theta}_{0,\text{s}}\right\|_{\ell_2} = \sqrt{\rho}r_{\text{s}}\,,$$

$$\left\|(\boldsymbol{U}^\perp)^\mathsf{T}\boldsymbol{\theta}_0\right\|_{\ell_2}^2 = \|\boldsymbol{\theta}_0\|_{\ell_2}^2 - \left\|\boldsymbol{U}^\mathsf{T}\boldsymbol{\theta}_0\right\|_{\ell_2}^2 = (1-\rho)r_{\text{s}}^2 + r_{\text{ns}}^2\,.$$

We next proceed to compute the risk of estimator in terms of $\boldsymbol{\gamma}$, $\gamma_0$. We use equation (B.6), which for the min-norm estimator with no look-alike clustering, reduces to

$$\text{Risk}(\widehat{\boldsymbol{\theta}}) = \sigma^2 + \gamma_0^2 + \boldsymbol{\gamma}^\mathsf{T}(\boldsymbol{I} + \mu^2\text{diag}(\boldsymbol{\pi}))\boldsymbol{\gamma}\,. \qquad \text{(B.18)}$$

This concludes the proof. Note that in the theorem statement we made the change of variables $\gamma_0 \to \tilde{\gamma}_0$ and $\boldsymbol{\gamma} \to \tilde{\boldsymbol{\alpha}}$, for an easier comparison with the risk of look-alike estimator.)

## B.4 Proof of Proposition 3.4

Consider singular value decompositions $\boldsymbol{X}_L = \boldsymbol{U}\boldsymbol{\Sigma}\boldsymbol{V}^T$ and $\tilde{\boldsymbol{X}}_L = \widetilde{\boldsymbol{U}}\widetilde{\boldsymbol{\Sigma}}\widetilde{\boldsymbol{V}}^T$. We then can write the estimators $\widetilde{\boldsymbol{\theta}}_L$ and $\widehat{\boldsymbol{\theta}}_L$ as follows:

$$\widehat{\boldsymbol{\theta}}_L = \boldsymbol{U}\boldsymbol{\Sigma}^{-1}\boldsymbol{V}^\mathsf{T}\boldsymbol{y}, \quad \widetilde{\boldsymbol{\theta}}_L = \widetilde{\boldsymbol{U}}\widetilde{\boldsymbol{\Sigma}}^{-1}\widetilde{\boldsymbol{V}}^\mathsf{T}\boldsymbol{y}.$$

We first bound $\|\widehat{\boldsymbol{\theta}}_L - \widetilde{\boldsymbol{\theta}}_L\|$. We write

$$\|\widehat{\boldsymbol{\theta}}_L - \widetilde{\boldsymbol{\theta}}_L\| \le \|\boldsymbol{U}\boldsymbol{\Sigma}^{-1}\boldsymbol{V}^\mathsf{T} - \widetilde{\boldsymbol{U}}\widetilde{\boldsymbol{\Sigma}}^{-1}\widetilde{\boldsymbol{V}}^\mathsf{T}\|\|\boldsymbol{y}\|. \tag{B.19}$$

We have

$$\|\boldsymbol{y}\| = \|\boldsymbol{X}^\mathsf{T}\boldsymbol{\theta}_0 + \varepsilon\| = \|\boldsymbol{\Lambda}^\mathsf{T}\boldsymbol{M}^\mathsf{T}\boldsymbol{\theta}_0 + \boldsymbol{Z}^\mathsf{T}\boldsymbol{\theta}_0 + \varepsilon\|.$$

Note that $\boldsymbol{Z}^\mathsf{T}\boldsymbol{\theta}_0 + \varepsilon \overset{(d)}{=} \sqrt{\|\boldsymbol{\theta}_0\|^2 + \sigma^2}\boldsymbol{g}$ where $\boldsymbol{g} \sim \mathsf{N}(0, I_n)$. In addition,

$$
\begin{aligned}
\frac{1}{n}\|\boldsymbol{\Lambda}^\mathsf{T}\boldsymbol{M}^\mathsf{T}\boldsymbol{\theta}_0\|^2 &= \frac{1}{n}\boldsymbol{\theta}_0^\mathsf{T}\boldsymbol{M}\boldsymbol{\Lambda}\boldsymbol{\Lambda}^\mathsf{T}\boldsymbol{M}^\mathsf{T}\boldsymbol{\theta}_0 \\
&= \boldsymbol{\theta}_0^\mathsf{T}\boldsymbol{M}\mathrm{diag}(\tfrac{n_1}{n}, \ldots, \tfrac{n_k}{n})\boldsymbol{M}^\mathsf{T}\boldsymbol{\theta}_0 \overset{p}{\to} \boldsymbol{\theta}_0^\mathsf{T}\boldsymbol{M}\mathrm{diag}(\pi_1, \ldots, \pi_k)\boldsymbol{M}^\mathsf{T}\boldsymbol{\theta}_0.
\end{aligned}
$$

Therefore by using concentration of Lipschitz functions of Gaussian vectors, we get

$$\frac{1}{\sqrt{n}}\|\boldsymbol{y}\| \overset{p}{\to} \sqrt{\boldsymbol{\theta}_0^\mathsf{T}\boldsymbol{M}\mathrm{diag}(\boldsymbol{\pi})\boldsymbol{M}^\mathsf{T}\boldsymbol{\theta}_0 + \|\boldsymbol{\theta}_0\|^2 + \sigma^2}.$$

This shows that

$$\frac{1}{\sqrt{n}}\|\boldsymbol{y}\| \to C \le \sqrt{(\mu + 1)(r_\mathrm{s}^2 + r_\mathrm{ns}^2) + \sigma^2}. \tag{B.20}$$

We next use the result of [28, Theorem 3.3], by which we obtain

$$\|\boldsymbol{U}\boldsymbol{\Sigma}^{-1}\boldsymbol{V}^\mathsf{T} - \widetilde{\boldsymbol{U}}\widetilde{\boldsymbol{\Sigma}}^{-1}\widetilde{\boldsymbol{V}}^\mathsf{T}\| \le \frac{1+\sqrt{5}}{2}\max\left(\frac{1}{\sigma_{\min}(\boldsymbol{\Sigma})^2}, \frac{1}{\sigma_{\min}(\widetilde{\boldsymbol{\Sigma}})^2}\right)\|\boldsymbol{U}\boldsymbol{\Sigma}\boldsymbol{V}^\mathsf{T} - \widetilde{\boldsymbol{U}}\widetilde{\boldsymbol{\Sigma}}\widetilde{\boldsymbol{V}}^\mathsf{T}\|. \tag{B.21}$$

Note that

$$\|\boldsymbol{U}\boldsymbol{\Sigma}\boldsymbol{V}^\mathsf{T} - \widetilde{\boldsymbol{U}}\widetilde{\boldsymbol{\Sigma}}\widetilde{\boldsymbol{V}}^\mathsf{T}\| = \|\boldsymbol{X}_L - \tilde{\boldsymbol{X}}_L\| = \|\boldsymbol{M}_\mathrm{s}\boldsymbol{\Lambda} - \widetilde{\boldsymbol{M}}_s\widetilde{\boldsymbol{\Lambda}}\| \le \delta\sqrt{n}, \tag{B.22}$$

by the assumption of the theorem statement. We next lower bound $\sigma_{\min}(\boldsymbol{\Sigma}) = \sigma_{\min}(\boldsymbol{X}_L)$. Recall that $\boldsymbol{X}_L^\mathsf{T} = (\boldsymbol{M}\boldsymbol{\Lambda})^\mathsf{T} + [\boldsymbol{0}_{n\times p}, \boldsymbol{Z}_{n\times(d-p)}]$, with $\boldsymbol{Z}$ having i.i.d $\mathsf{N}(0,1)$ entries.

Next suppose that Condition $(i)$ holds true, namely $\delta < \sqrt{1 - (\psi_d - \psi_p)} - \sqrt{\psi_d - \psi_p}$, with $\psi_d - \psi_p < 0.5$. Using the result of [31, Theorem 2.1], we have with probability at least $1 - n^{-1}$,

$$\sigma_{\min}(\boldsymbol{X}_L) \ge \sqrt{n}\left(\sqrt{\psi_d - \psi_p - 1} - 1 - \sqrt{\frac{2\log n}{n}}\right).$$

Furthermore,

$$
\begin{aligned}
\sigma_{\min}(\tilde{\boldsymbol{X}}_L) &\ge \sigma_{\min}(\boldsymbol{X}_L) - \|\boldsymbol{X}_L - \tilde{\boldsymbol{X}}_L\| \\
&\ge \sqrt{n}\left(\sqrt{1 - (\psi_d - \psi_p)} - \sqrt{\psi_d - \psi_p} - \sqrt{\frac{2\log n}{n}} - \delta\right) \\
&\ge c'\sqrt{n}\left(\sqrt{1 - (\psi_d - \psi_p)} - \sqrt{\psi_d - \psi_p}\right),
\end{aligned}
$$

using the assumption on the estimation error rate $\delta$. Therefore, using the above bound along with (B.22) in (B.21) we get

$$\|\boldsymbol{U}\boldsymbol{\Sigma}^{-1}\boldsymbol{V}^\mathsf{T} - \widetilde{\boldsymbol{U}}\widetilde{\boldsymbol{\Sigma}}^{-1}\widetilde{\boldsymbol{V}}^\mathsf{T}\| \le \frac{1+\sqrt{5}}{2c'^2}\frac{1}{\sqrt{n}\left(\sqrt{1 - (\psi_d - \psi_p)} - \sqrt{\psi_d - \psi_p}\right)^2}\delta.$$

Combining the above bound with (B.20), we get

$$\|\widehat{\boldsymbol{\theta}}_L - \widetilde{\boldsymbol{\theta}}_L\| \le \frac{1+\sqrt{5}}{2c'^2} \frac{C}{\left(\sqrt{1-(\psi_d-\psi_p)} - \sqrt{\psi_d-\psi_p}\right)^2} \delta \,. \tag{B.23}$$

We next note that by triangle inequality, the above bound implies that

$$\|\widetilde{\boldsymbol{\theta}}_L - \boldsymbol{\theta}_0\| - \|\widehat{\boldsymbol{\theta}}_L - \boldsymbol{\theta}_0\| \le \|\widehat{\boldsymbol{\theta}}_L - \widetilde{\boldsymbol{\theta}}_L\| = O(\delta) \,.$$

Therefore, by invoking Lemma 2.1, we obtain the desired result on $\text{Risk}(\widetilde{\boldsymbol{\theta}}_L)$.

Next suppose that Condition $(ii)$ holds, namely $\delta < \sqrt{\psi_d - \psi_p - 1} - 1$ with $\psi_d - \psi_p > 2$. Using the result of [31, Theorem 2.1] for $\boldsymbol{X}^\mathsf{T}$, we have with probability at least $1 - n^{-1}$,

$$\sigma_{\min}(\boldsymbol{X}_L) \ge \sqrt{n}\left(\sqrt{\psi_d - \psi_p - 1} - 1 - \sqrt{\frac{2\log n}{n}}\right) \,.$$

By following a similar argument we prove the claim under Condition $(ii)$.

## B.5 Proof of Theorem 5.1

We use Theorem 3.3 (b) to characterize $\text{Risk}(\widehat{\boldsymbol{\theta}})$ in the regime of $\psi_d \ge 1$. Specializing to the case of balanced cluster priors, the risk depends on $\tilde{\boldsymbol{\alpha}}$ only through its norm $\tilde{\alpha} := \|\tilde{\boldsymbol{\alpha}}\|_{\ell_2}$, and is given by

$$\text{Risk}(\widehat{\boldsymbol{\theta}}) \overset{\mathcal{P}}{\to} \sigma^2 + \tilde{\gamma}_0^2 + \left(\frac{\mu^2}{k} + 1\right)\tilde{\alpha}^2$$

$$= \frac{\psi_d}{\psi_d - 1}\left(\sigma^2 + \left(\frac{\mu^2}{k} + 1\right)\tilde{\alpha}^2\right) + \left(1 - \frac{1}{\psi_d}\right)((1-\rho)r_\text{s}^2 + r_\text{ns}^2),$$

with

$$\tilde{\alpha} = \left(1 + \frac{\frac{\mu^2}{k} + 1}{\psi_d - 1}\right)^{-1}\sqrt{\rho}r_\text{s} \,.$$

In addition, by Theorem 3.1 we have

$$\text{Risk}(\widehat{\boldsymbol{\theta}}_L) \overset{\mathcal{P}}{\to} \frac{\sigma^2 + r_\text{s}^2}{1 - \psi_d + \psi_p} - \rho r_\text{s}^2 \,.$$

Note that $\text{Risk}(\widehat{\boldsymbol{\theta}}_L)$ in this regime does not depend on $\mu^2/k$. Also, it is easy to verify that $\text{Risk}(\widehat{\boldsymbol{\theta}})$ is decreasing in $\mu^2/k$. Therefore the gain $\Delta$ is decreasing in $\mu^2/k$.

Also observe that $\text{Risk}(\widehat{\boldsymbol{\theta}})$ is increasing in $r_\text{ns}$, while $\text{Risk}(\widehat{\boldsymbol{\theta}}_L)$ does not depend on $r_\text{ns}$. Therefore, the gain $\Delta$ is increasing in $r_\text{ns}$.

To understand the dependence of $\Delta$ on $\rho$, we write

$$\Delta - 1 = \frac{\text{Risk}(\widehat{\boldsymbol{\theta}})}{\text{Risk}(\widehat{\boldsymbol{\theta}}_L)} - 1$$

$$= \frac{\frac{\psi_d}{\psi_d - 1}\left(\sigma^2 + \left(\frac{\mu^2}{k} + 1\right)\left(1 + \frac{\mu^2/k+1}{\psi_d-1}\right)^{-2}\rho r_\text{s}^2\right) + \left(1 - \frac{1}{\psi_d}\right)((1-\rho)r_\text{s}^2 + r_\text{ns}^2)}{\frac{\sigma^2 + r_\text{s}^2}{1 - \psi_d + \psi_p} - \rho r_\text{s}^2} - 1$$

$$= \frac{\frac{\psi_d}{\psi_d - 1}\left(\sigma^2 + \left(\frac{\mu^2}{k} + 1\right)\left(1 + \frac{\mu^2/k+1}{\psi_d-1}\right)^{-2}\rho r_\text{s}^2\right) + \left(1 - \frac{1}{\psi_d}\right)(r_\text{s}^2 + r_\text{ns}^2) - \frac{\sigma^2 + r_\text{s}^2}{1 - \psi_d + \psi_p} + \frac{\rho r_\text{s}^2}{\psi_d}}{\frac{\sigma^2 + r_\text{s}^2}{1 - \psi_d + \psi_p} - \rho r_\text{s}^2}$$

As we see the numerator is increasing in $\rho$ and denominator is decreasing in $\rho$, which implies that the gain $\Delta$ is increasing in $\rho$.

We next show that $\Delta \geq 1$ if condition (5.1) holds. Since $\Delta$ is decreasing in $\mu^2/k$ and increasing in $\rho$, it suffices to show the claim assuming $\mu^2/k \to \infty$ and $\rho = 0$. In this case we have $\left(\frac{\mu^2}{k} + 1\right)\tilde{\alpha}^2 \to 0$ and so

$$
\Delta \to \frac{\frac{\sigma^2 \psi_d}{\psi_d - 1} + \left(1 - \frac{1}{\psi_d}\right)\left(r_{\mathrm{s}}^2 + r_{\mathrm{ns}}^2\right)}{\frac{\sigma^2 + r_{\mathrm{s}}^2}{1 - \psi_d + \psi_p}}
$$

$$
\geq \frac{\frac{\sigma^2 \psi_d}{\psi_d - 1} + \left(1 - \frac{1}{\psi_d}\right)r_{\mathrm{s}}^2}{\frac{\sigma^2 + r_{\mathrm{s}}^2}{1 - \psi_d + \psi_p}}
$$

$$
= \frac{\frac{\psi_d}{\psi_d - 1} + \left(1 - \frac{1}{\psi_d}\right)\mathrm{SNR}^2}{\frac{1 + \mathrm{SNR}^2}{1 - \psi_d + \psi_p}} \geq 1,
$$

where the last step follows from condition (5.1).

