$$
\boldsymbol{X}_L = \begin{bmatrix} \boldsymbol{M}_\text{s}\boldsymbol{\Lambda} \\ \boldsymbol{M}_\text{ns}\boldsymbol{\Lambda} + \boldsymbol{Z}_\text{ns} \end{bmatrix},
$$

and therefore by substituting for $\boldsymbol{y}$, $\boldsymbol{X}$, and $\boldsymbol{X}_L$, we get

$$
\begin{aligned}
\frac{1}{2n} \left\| \boldsymbol{y} - \boldsymbol{X}_L^\mathsf{T}\boldsymbol{\theta} \right\|_{\ell_2}^2 &= \frac{1}{2n} \left\| \varepsilon + \boldsymbol{X}^\mathsf{T}\boldsymbol{\theta}_0 - \boldsymbol{X}_L^\mathsf{T}\boldsymbol{\theta} \right\|_{\ell_2}^2 \\
&= \frac{1}{2n} \left\| \varepsilon + \boldsymbol{\Lambda}^\mathsf{T}\boldsymbol{M}_\text{s}^\mathsf{T}(\boldsymbol{\theta}_{0,\text{s}} - \boldsymbol{\theta}_\text{s}) + \boldsymbol{Z}_\text{s}^\mathsf{T}\boldsymbol{\theta}_{0,\text{s}} + (\boldsymbol{\Lambda}^\mathsf{T}\boldsymbol{M}_\text{ns}^\mathsf{T} + \boldsymbol{Z}_{ns}^\mathsf{T})(\boldsymbol{\theta}_{0,\text{ns}} - \boldsymbol{\theta}_\text{ns}) \right\|_{\ell_2}^2 \,.
\end{aligned}
$$

We define the primary optimization loss as follows:

$$
\mathcal{L}_{PO}(\boldsymbol{\theta}_\text{s}, \boldsymbol{\theta}_\text{ns}) := \frac{1}{2n} \left\| \varepsilon + \boldsymbol{\Lambda}^\mathsf{T}\boldsymbol{M}_\text{s}^\mathsf{T}(\boldsymbol{\theta}_{0,\text{s}} - \boldsymbol{\theta}_\text{s}) + \boldsymbol{Z}_\text{s}^\mathsf{T}\boldsymbol{\theta}_{0,\text{s}} + (\boldsymbol{\Lambda}^\mathsf{T}\boldsymbol{M}_\text{ns}^\mathsf{T} + \boldsymbol{Z}_{ns}^\mathsf{T})(\boldsymbol{\theta}_{0,\text{ns}} - \boldsymbol{\theta}_\text{ns}) \right\|_{\ell_2}^2 + \lambda \|\boldsymbol{\theta}_\text{s}\|_{\ell_2}^2 + \lambda \|\boldsymbol{\theta}_\text{ns}\|_{\ell_2}^2
$$

We continue by deriving the auxiliary optimization (AO) problem. By duality, we have

$$
\begin{aligned}
\mathcal{L}_{PO}(\boldsymbol{\theta}_\text{s}, \boldsymbol{\theta}_\text{ns}) &= \max_{\boldsymbol{v}} \frac{1}{n} \left( \boldsymbol{v}^\mathsf{T}\varepsilon + \boldsymbol{v}^\mathsf{T}\boldsymbol{\Lambda}^\mathsf{T}\boldsymbol{M}_\text{s}^\mathsf{T}(\boldsymbol{\theta}_{0,\text{s}} - \boldsymbol{\theta}_\text{s}) + \boldsymbol{v}^\mathsf{T}\boldsymbol{Z}_\text{s}^\mathsf{T}\boldsymbol{\theta}_{0,\text{s}} + \boldsymbol{v}^\mathsf{T}(\boldsymbol{\Lambda}^\mathsf{T}\boldsymbol{M}_\text{ns}^\mathsf{T} + \boldsymbol{Z}_{ns}^\mathsf{T})(\boldsymbol{\theta}_{0,\text{ns}} - \boldsymbol{\theta}_\text{ns}) - \frac{\|\boldsymbol{v}\|_{\ell_2}^2}{2} \right) \\
&\quad + \lambda \|\boldsymbol{\theta}_\text{s}\|_{\ell_2}^2 + \lambda \|\boldsymbol{\theta}_\text{