# OpenReview forum: "Anonymous Learning via Look-Alike Clustering: A Precise Analysis of Model Generalization"
_NeurIPS.cc/2023/Conference — NeurIPS 2023 poster_

### Official Review · Reviewer_Kp2o · 2023-07-04

**Soundness:** 3 good
**Presentation:** 3 good
**Contribution:** 2 fair
**Rating:** 6
**Confidence:** 3

**Summary:**

This work considers linear regression on data $(x_i, y_i)_{i=1}^n$ where each $x_i \in \mathbb{R}^d$ is sampled from one of $k$ Gaussian clusters, each with probability $\pi_k$, and $y_i = x_i ^T \theta_0 + \varepsilon_i$ with $\varepsilon_i \sim N(0, \sigma^2)$ representing noise. The goal of this paper is to characterize and compare the out-of-sample expected squared error of the minimum 2-norm regression predicter with that of the so called ``look-alike" predicter. The look-alike predicter is the minimum 2-norm predicter after one replaces the first $p$ features of each point with the first $p$ features of their respective cluster average: to motivate this consider the first $p$ features as sensitive in some manner, aggregation therefore provides a degree of anonymization for each data point. The min-norm and look-alike predicter are then analyzed in the asymptotic regime $d,p,n \rightarrow \infty$ with $d/n \rightarrow \varphi_d$ and $p/n \rightarrow \varphi_p$. The min-norm and look-alike estimators are said to be in the underparameterized regime if $\varphi_d\leq 1$ and $\varphi_d - \varphi_p \leq 1$ respectively, and are otherwise considered overparameterized. In addition, if $\theta_s$ denotes the first $p$ (sensitive) entries of $\theta_0$, then the Signal to Noise Ratio SNR is defined as $||\theta_s || / \sigma$. This work identifies a number of scenarios in which the look-alike predicter outperforms the min-norm predicter, in particular when the SNR is low then the underparmaterized look-alike predicter outperforms the min-norm predicter.

**Strengths:**

This paper illustrates, albeit in a specific setting, that anonymization by aggregating sensitive features does not necessarily lead to a drop in the performance of the predictor, and can in certain settings actually lead to improvements. The results appear technically sound and are based on a nice use of the Convex-Gaussian Minimax Theorem. I did not go through all of the appendix or re-derive any of the results line-by-line. Synthetic data is used to support and validate the results numerically. The paper is also well written and structured. I am not an expert in this space so cannot comment really on the novelty of the work.

**Weaknesses:**

The biggest weaknesses to my mind is perhaps the specificity of the data model, both in terms of the data model. Little discussion as to the feasibility of extending the takeaways to more general settings is provided. As a small note I also think greater care is perhaps required when discussing the informal notion of user privacy used here versus more formal notions, e.g., differential privacy, which actually provide privacy guarantees.

**Questions:**

1. Can you comment as to the potential avenues and barriers to relaxing your results from asymptotic to big-O?

2. Can you comment as to the potential for your results to be extended to more general data distributions?

3. In Figure 3a) when $\varphi_p=0.9$ then $p=d=0.9n$ and all features are sensitive. If in addition the SNR is $0$, then unless I am mistaken this implies $\theta_0 = 0 $ and therefore $y_i = \varepsilon_i$, i.e., the target is independent of the input features and is random noise. It seems curious then that one classifier has a better generalization than another in this setting, could you comment?

3. Have you observed the conclusions of your results to actually hold on any real-world data? In particular have you identified any real world data problems where you actually see the look-alike estimator out-perform the min-norm estimator?

4. I wonder if the proportional regime is the most appropriate to consider here for actual applications versus say fixed $d$ and $p$, for instance when considering financial or health records in a country. Could you comment on the how your results might change in this setting?

**Limitations:**

A section discussing limitations of the work is lacking. I cannot foresee any negative societal outcomes from this work. Although it seems only fairly basic numerics are presented a link to the code used to generate the plots is lacking and should probably be included.

---

> ### Author Rebuttal · Authors · 2023-08-09
>
> Thank you for taking the time to read and review our paper!
>
> ## General comments on weaknesses:
>
> 1) **Limitations of the model**: We expect that the virtue of look-alike modeling on generalization to apply in a broad range of settings. The high-level intuition is that the look-alike modeling acts as a regularizer and can improve the generalization by reducing the variance, an observation that is not limited to linear regression. To support it empirically, in our **global response** we provide additional experiments for a nonlinear data model and empirically observe a similar phenomena (i.e., look-alike modeling achieves better generalization over the vanilla estimator at lower SNR). In terms of theory, a similar approach based on CGMT can be used to handle other loss functions, albeit leading to a much more complicated formula for the generalization, which is less transparent in showing the message. A limitation in our analysis is though the distribution of features, as the CGMT strongly relies on gaussianity of the features.
>
>
> 2) **Privacy measures**: We will add a discussion on other privacy measures, including differential privacy, aggregate learning and k-anonymity. Note that in our setting if the minimum size cluster is k, then after look-alike modeling we have k-anonymity in the sense that for any individual there are at least k-1 other users with the same sensitive features.
>
> ## Response to questions:
>
> 1) The proportional regime (where n,d, p are of the same order) studied in the paper is indeed much more challenging than the population regime where the sample size $n\gg d,p$. In the population regime, the variance of the estimator goes to zero and we are essentially left with a deterministic estimator. However, as we mentioned in the paper, the population regime is not relevant to practical situations as many ML systems (in particular neural nets) have a humongous number of parameters, on the order of number of training samples.
>
> 2) Please see our response above (#1 in weakness)
>
> 3) Yes, the target in this case is just random noise, but note that the two estimators are using different features to fit to this noise vector (one uses individual features and one uses cluster centers), so you will naturally have different estimators and so different risks.
>
> 4) We have indeed observed this phenomena on a click-through-rate dataset of users on a pool of ads. However, due to privacy of users we are not permitted to share that data.
>
> 5) Although our theory is for an asymptotic regime, as we show in the simulations, when $n,d,p$ are only a few hundreds we already see a great match between theory and simulations.

---

> > ### Comment · Reviewer_Kp2o · 2023-08-14
> >
> > Thanks for answering my questions, on balance I will keep my current score.

---

### Official Review · Reviewer_PpUR · 2023-07-06

**Soundness:** 2 fair
**Presentation:** 2 fair
**Contribution:** 3 good
**Rating:** 6
**Confidence:** 2

**Summary:**

In this paper the authors propose a look-alike clustering technique that replaces sensitive feature of individuals with the cluster’s average value – the cluster in which that individual belongs to. The authors provide precise analysis of how replacing sensitive features with cluster center value affect the generalization of the linear regression model by comparing under-parameterized and over-parameterized regimes in the asymptotic setting. Numerical experiments are provided that matches the asymptotic theory.

**Strengths:**

1.	Precise analysis of how replacing sensitive features with cluster center value affect the generalization of the linear regression model.
2.	Expression of risk in the under-parameterized and over-parameterized regimes and analysis of scenarios when the proposed method improves generalization by achieving lower risk while ensuring privacy.


**Weaknesses:**

1.	While the introduction suggests that the developed theory clearly demonstrates the role of cluster size and number of clusters in controlling generalization error none of the theory statements presented in the main paper explicitly shows that dependence. While looking at the supplementary material, it is clear that after simple algebraic manipulation such effect can be seen (e.g., in part (b) of Theorem 3.3) it may be a better idea to reorganize the theory statement where such dependence is clearly visible.
2.	In theorem 3.1 what is $\rho$? While later at somewhere else it is mentioned that it captures the alignment of the model with the left singular vectors of the cluster centers can it be interpreted as $U^T_s\theta_{0,s}=\sqrt{\rho}\theta_{0,s}$ which yields $|| U^T_s\theta_{0,s}||=\sqrt{\rho}r_s$?
3.	For case 1 in section 5, $\psi_d=0.9$. If we look at figure 3(a), among various values of $\psi_p$, $\psi_p=0.9$ tolerates largest SNR while still ensuring $\Delta>1 (\log (\Delta) >0)$. However, since $\psi_d=0.9, \psi_p=0.9$ means all the features are sensitive and yet it tolerates the largest SNR and improves generalization. Does that mean, irrespective of sensitiveness, it is always a good strategy to replace all features by their cluster centers to achieve better generalization?
4.	If we compare Figure 3(a) and 3(b) and look at what SNR values the red curve hits zero for $\rho=0.3$ and $\psi_p=0.5$, we get very different values. On Fig 3(a), this SNR value is a little less than 2 while on Fig 3(b), this SNR value is close to 2.5. Since both axes in Fig 3(a), 3(b) are identical, why is this discrepancy?
5.	In figure 2, the six graphs are organized in a 2x3 window without clear “left” and “right” panels as mentioned in the figure caption. Reorganizing them in 2x3 window will have two panels and will match the figure caption.
6.	Few typos in the statement of Proposition 3.
       a.	 $\tilde{X}^T$ should be $\tilde{X}^T_L$
       b.	  It seems $\delta_n$ and $\delta$ represent same thing (cluster estimation error rate). This needs to be fixed.
7.	In line 200, “underparameterized” should be “overparameterized”.


**Questions:**

1.	See weakness.

**Limitations:**

Limitations are not clearly mentioned. To establish concrete expression of the risk, Assumption 2 is very simplistic or the isotropic Gaussians in the GMM is very simplistic. It may be a good idea to add a specific section/subsection mentioning the limitations.

---

> ### Author Rebuttal · Authors · 2023-08-09
>
> Thank you for taking the time to read and review our paper!
> ## General comments on weaknesses:
> 1) We respectfully disagree with your comment. Our theory indeed captures the effect of cluster size and the number of clusters as well as other factors precisely in the studied asymptotic regime. However, an intriguing corollary of our theory is that in the so-called underparamterized regime, those two factors (number of size of clusters) do not impact the generalization (cf. Theorem 3.1), while in the overparamterized regime they do (cf Theorem 3.2 and 3.3). Maybe that is the source of your confusion. We have highlighted this point in lines 106-111 and 191-192.
>
> 2) Definition of $\rho$ is just stated in the theorem itself (see line 185). Please recheck.
>
> 3) Not necessarily. Recall that $r_s := ||\theta_{0,s}||$ is the norm of model components restricted to sensitive features. So if you continue converting nonsensitive features to the sensitive ones, $r_s$ and so the SNR goes up. In other words, when it gets to comparing gain, you will be comparing different curves at different SNRs.
>
> 4) It seems that you are comparing wrong curves. Note that $\rho = 0.3$, $\psi = 0.5$ corresponds to the **green** curve in Fig 3(a) and to the **red** curve in Fig 3(b).
>
> 5) Thanks for raising it. It happened as we were dealing with the space constraint. We will fix the arrangement of panels.
>
> 6,7)  Thanks! We will fix these typos.

---

> > ### Comment · Reviewer_PpUR · 2023-08-16
> >
> > Thank you for clarifying some of my questions and confusions. Reading the reviews posted by other reviewers and the related discussion, it seems that I misjudged the merit of this paper. Therefore, I raise my score to 6.

---

### Official Review · Reviewer_KLzk · 2023-07-06

**Soundness:** 3 good
**Presentation:** 3 good
**Contribution:** 2 fair
**Rating:** 6
**Confidence:** 4

**Summary:**

This work presents a generalization analysis for look-alike clustering. In this type of clustering, the features in a model are divided into two groups: sensitive and non-sensitive, and the values of the sensitive features are replaced by the mean of the cluster to ensure K-anonymity (if the size of the cluster is K). They provide an analysis of generalization in the under-parameterized and over-parametrized regimes.


**Strengths:**

The paper brings in generalization analysis for look-alike clustering, i.e., clustering under the assumptions of some features being held fixed at their mean. They provide some simulations as well, showing the under-parametrized and over-parametrized regimes.

**Weaknesses:**

It would be good to provide more intuitive insights on how different factors affect the generalization, e.g., the number of sensitive and nonsensitive features. My intuition is that as the number of sensitive features increases, the generalization should get worse until it fails as they approach the total number of features?

I do not understand this assumption: "We focus on an asymptotic regime where the size of the training set grows in proportion to the features dimension." For a complete picture, it is best to keep them as separate variables, and if required, consider a substitution at the end in a corollary or something. This assumption takes away important insights related to the relationship between number of features and size of training set.

Experiments are mostly on synthetic data but it may be okay since the focus is mostly theoretical.

Clustering assumptions are based on linear models.

**Questions:**

Clarify what happens if number of features are not equal to number of samples?

How do number of sensitive features affect generalization?

---

> ### Author Rebuttal · Authors · 2023-08-09
>
> Thank you for taking the time to read and review our paper!
> ## General comments on weaknesses:
> 1) The effect of the number of sensitive features on the generalization is indeed more complicated. For example, in the underparamterized regime as shown in Theorem 3.1, if we fix $n,d$ and increase $p$ (so $\psi_d = d/n$ is fixed and $\psi_p = p/n$ increases) the denominator in the risk increases, so does $r_s$ in the numerator (recall that $r_s = ||\theta_{0,s}||$ represents the norm of model component restricted to sensitive features). In addition, $\rho$ can also change as it measures the correlation between $\theta_{0,s}$ (the model component over sensitive features) and the cluster centers. The overall effect on the risk depends on the interplay between these factors. The virtue of our precise theory is to shed light on such intriguing effects. The reason that your intuition does not give the complete picture is at core due to a bias-variance tradeoff.  As we argue after Theorem 5.1, look-alike modeling acts as a regularizer. As we increase the number of sensitive features, the impact of such regularization becomes stronger. It induces bias but also decreases the variance of the estimators. Depending on the interplay between them, we can have a positive or negative impact on the generalization.
>
> 2) You have a mis-understanding about the proportion regime. It says that the sample size and the features dimension are “proportional”, i.e. they are of the same “order” as the sample size grows to infinity (mathematically $d/n \to \psi_d$ and $p/n \to\psi_p$ as $n\to\infty$ for arbitrary bounded constants $\psi_d$ and $\psi_p$).Therefore, $n$, $d$, $p$ are separate variables (e.g., d = 1000n or p = 100 n, and these constants can be arbitrary).
> This is in contrast to the “population regime” where the sample size is “orderwise” larger than the features dimension (i., n/d \to \infty). The population regime is of course much easier to analyze, however it does not capture many of the interesting phenomena happening in the proportional regime. Moreover, as discussed in the paper, with rise of overparamterized regime, the proportional regime is more relevant in practice.
>
> 3) Since our goal have been on deriving a precise characterization of generalization performance of look-alike modling, we devoted our numerical experiments to synthetic data to corroborate our theory and show the perfect match between simulations and the proposed theory.
>
> 4) Regarding the linear model and the limitations, please see our **global response** to all reviewers where we supplement an experiment with a non-linear data model and empirically show that a similar phenomena (better generalization of the look-alike modeling at low SNR) applies to this setting as well.
>
> ## Response to questions:
>
> 1) Please see our response above to it. We do not make such assumption. The proportional regime assumes that the number of features and the number of samples are at the same scale (their ratio can be any arbitrary bounded constant.)
>
> 2) Please see our response above (item #1 in weaknesses).

---

> > ### Comment · Reviewer_KLzk · 2023-08-14
> >
> > Increasing my score by 1 (5 to 6)
> > My understanding of the assumption was correct, but what I was asking was why this is a good assumption.
> > In the final version, please include a remark discussing this assumption. It would be good to also find other theory papers where this proportional regime has been considered.

---

> > > ### Author Response · Authors · 2023-08-16
> > >
> > > Thank you for your comments and also for raising your score. We will include a remark in the final version to further explain this asymptotic regime, its relevance as well as several additional work which has considered this proportional regime.

---

### Official Review · Reviewer_gwNk · 2023-08-01

**Soundness:** 3 good
**Presentation:** 3 good
**Contribution:** 3 good
**Rating:** 6
**Confidence:** 4

**Summary:**

This paper extends look-alike clustering to anonymize sensitive features of data points by replacing them with cluster means. It provides a theoretical analysis of the generalization error of models (linear regression estimator) trained using the anonymized sensitive features in the asymptotic regime. The paper also shows that anonymizing sensitive features through look-alike clustering can improve model generalization by acting as a regularization and mitigating overfitting when the signal-to-noise ratio (ratio of the energy of the model on sensitive features to the response noise) is below a certain threshold under different parametrization regimes.


**Strengths:**

1. The paper studies an interesting area of anonymizing private/sensitive user information, drawing on recent research that examines the memorization of training data samples.

2. The theoretical analysis presented in the paper is intriguing as it explores scenarios where learning from look-alike clustering-based anonymized data can potentially improve model generalization.

3. The paper is well-written with the majority of sections being clear and coherent.


**Weaknesses:**

1. The current draft lacks coverage of related works on other data anonymization methods. It would be helpful for the readers if the authors included a literature survey in either the main paper or the appendix.

2. The paper lacks comprehensive empirical experiments on real-world datasets (of any scale).

3. Limitations of the theoretical analysis presented in the paper are also missing.

4. Perturbation analysis about the upper bound of the look-alike estimator's risk (Proposition 3.4) is not covered in the over-parametrized regime.


**Questions:**

1. Please refer to the comments in the weaknesses section.

2. Besides the generalization bounds, do the authors have any insights about the DP guarantees of their proposed look-alike anonymization approach?

Minor recommendations:

1.  The authors should specify that proposition 3.4 holds true for the look-alike estimator in the under-parametrized regime.

Reference for k-anonymity :

1.  Sweeney, Latanya. "k-anonymity: A model for protecting privacy." International journal of uncertainty, fuzziness and knowledge-based systems 10.05 (2002): 557-570.

Related works covering memorization of sensitive information in neural networks and can provide valuable insights into the problem motivation:
1. Song, Congzheng, and Vitaly Shmatikov. "Overlearning reveals sensitive attributes." arXiv preprint arXiv:1905.11742 (2019).

2. Feldman, Vitaly, and Chiyuan Zhang. "What neural networks memorize and why: Discovering the long tail via influence estimation." Advances in Neural Information Processing Systems 33 (2020): 2881-2891.

3. Stephenson, Cory, et al. "On the geometry of generalization and memorization in deep neural networks." arXiv preprint arXiv:2105.14602 (2021).

4. Malekzadeh, Mohammad, Anastasia Borovykh, and Deniz Gündüz. "Honest-but-curious nets: Sensitive attributes of private inputs can be secretly coded into the classifiers' outputs." Proceedings of the 2021 ACM SIGSAC Conference on Computer and Communications Security. 2021.



**Limitations:**

I could not find a section addressing the limitations and negative societal impact of this work.

---

> ### Author Rebuttal · Authors · 2023-08-09
>
> Thank you for taking the time to read and review our paper!
>
> ## General comments on weaknesses:
> 1) We will add related work on other data anonymization methods, including differential privacy, k-anonymity and aggregate learning.
>
> 2) Since the focus has been developing on “precise characterization” of the effect of look-alike modeling on generalization, we devoted our numerical sections on thorough synthetic simulations to corroborate our theory in various data regimes. We will be happy to add a real-data experiment to show the high-level idea that look-alike modeling can improve generalization.
>
> 3) A limitation of the presented analysis is that it is focused on linear regression. Despite its simplicity though, it is rich enough to show the surprising phenomenon that look-alike clustering can reduce overfitting. Also as you see deriving a precise characterization of generalization, in the regime that sample size and features dimension grow in proportion is already challenging. That said, we expect the same phenomenon carries over to a broader problem setting.
>
> 4) The upper bound on the perturbation becomes vacuous unless $\psi_d-\psi_p \ge 0.5$. Recall that $\psi_d-\psi_p = (d-p)/n$. Still we can have $\psi_p = p/n$ and $\psi_d = d/n$ both be higher than one, but it needs the number of non-sensitive features $(d-p)$ be smaller than $n$. The assumption arises in the analysis as we need to compare the spectrum of the look-alike features and the spectrum of the raw individual features and the randomness of the non-sensitive features can lead to large deviations in the eigenvalues. However, please note that this proposition only makes an assumption on the perturbation norm, and NOT any specific estimation of clusters (so perturbations can be added in an adversarial way as far as it satisfies the norm constraint).
>
> ## Response to questions:
> 1) Please see our responses to weaknesses.
>
> 2) It is a very interesting question. In the current form the look-alike modeling is not DP, unless some sort of randomization (noise) is applied to the look-alike features. In ongoing work we are investigating the right way of doing it. The idea is to split the privacy budget,  part of it to learn clusters privately and part of it to randomize the look-alike features. The intuition is that if the cluster sizes are large enough, then this approach leads to a better privacy/generalization tradeoff compared to making the individual features DP from the onset. But it is an ongoing work and out of the scope of the current one.
> Minor comments: Many thanks for the pointers. We will add/discuss them in the revised version.

---

### Official Review · Reviewer_EnfZ · 2023-08-02

**Soundness:** 4 excellent
**Presentation:** 3 good
**Contribution:** 3 good
**Rating:** 7
**Confidence:** 3

**Summary:**

The paper studies linear regression where some coordinates of the covariate x are not revealed directly to the learner, and only a cluster-wise average is revealed for those coordinates. This change in the covariates x is called look-alike clustering and it has been used for protecting sensitive attributes in prior work. While it seems that with look-alike clustering, less information is provided to the learner, the paper shows that in certain regimes the performance of linear regression improves after look-alike clustering. As the authors explain, the reason behind this surprising phenomenon is that look-alike clustering can reduce overfitting.

More concretely, this paper assumes that the data points (x_i, y_i) are generated iid from a linear model y_i = < x_i, theta > + eps_i with mean-zero Gaussian noise eps_i. The distribution of the covariate x is a mixture of k Gaussian distributions each with its covariance matrix being the identity matrix. Look-alike clustering then corresponds to replacing a subset of the coordinates of each x_i by the coordinates of the mean of the Gaussian distribution that generates x_i. Those coordinates are called sensitive features.

For d dimensional x with p sensitive features, the authors study the asymptotic performance of linear regression with n data points as d, p, n all tending to infinity and the ratios d/n, p/n tending to constants. The number of clusters, k, is fixed, and performance is measured using the (population) mean squared error. The learner estimates the parameter theta in the linear model by solving least squares on the data points. When there are multiple solutions (e.g. when n < d) the learner chooses the one with the minimum l_2 norm $\\|\theta\\|_2$.

The authors show closed-form formulas for the asymptotic performance with and without look-alike clustering. This allows the authors to identify regimes when look-alike clustering improves the performance. They also included experimental results verifying the formulas. While the formulas are only shown to hold asymptotically, even when d, p, n are only a few hundreds the experimental results already closely match the formulas. The authors also extend the formulas to the case where we do not know the means of the k Gaussians when performing look-alike clustering and we instead use estimates for these means.

**Strengths:**

The authors chose an interesting problem to study: the effect of look-alike clustering on generalization. The exact formulas on the performance of linear regression established in this paper are technically challenging to obtain, for which the authors use tools such as the Convex Gaussian Minimax Theorem. The paper is well written. The proofs in the supplementary are easy to follow and they look correct and complete.

**Weaknesses:**

1. Unless I am missing some parts of the paper, the authors do not provide any (empirical) examples beyond linear regression where look-alike clustering is helpful for performance. It would be great to have such examples as they may suggest that the analysis in this paper could explain a general phenomenon appearing not just in linear regression. Currently, the impact of the paper is somewhat limited to the linear regression setting.

2. In proposition 3.4 where look-alike clustering uses estimated means of the Gaussians, a requirement seems to be that we can accurately estimate the Gaussian mean of every data point. When we do not know which cluster each point comes from, this seems quite challenging especially when the Gaussians are not well separated. It is important for the authors to provide a clear explanation on how to obtain such accurate estimates (perhaps using prior results on learning Gaussian mixtures).

3. Starting at Line 297, the authors provide an explanation for the improved generalization from look-alike clustering. Indeed, with the main results being presented using math formulas, a high-level explanation is much needed. However, I am not able to fully understand the authors' explanations at Line 297. The authors wrote that look-alike clustering drops the <z_s, theta_s> term, but my understanding is that the y in look-alike clustering is generated using the original x and thus still contains the <z_s, theta_s> term, which means that the authors' description is inaccurate. It is also unclear how the authors' explanation is related to regularization.

    Here is my own understanding of why look-alike clustering can improve generalization. The authors can comment on whether this makes sense. After look-alike clustering, each x_i, when restricted to the sensitive features, can only be chosen from k possible vectors, and thus the learned parameter theta, when restricted to the sensitive features, must be in the space spanned by the k vectors (since the learner picks the minimum norm theta). This low-dimension restriction is a form of regularization and can reduce overfitting. This does not explain why lower SNR leads to larger performance gain, which I feel is not explained well in the authors' explanation either.

**Questions:**

1. The explanation at Line 297 seems to be presented only for Case 2, but I don't see why this should be the case.
2. It seems that the assumed upper bound on delta at Line 237 could be negative, in which case the assumption cannot be satisfied. I would appreciate some comments on this.
3. It would be great if the authors can provide their feedback on the weaknesses session of the review.

Typos:

- Line 29: leaner -> learner

- Line 30: that that -> that

- Line 133: belong -> belongs

**Limitations:**

The authors adequately stated all the assumptions needed for obtaining their results. In terms of negative social impact, perhaps one question is whether look-alike clustering may lead to stereotyping each cluster (e.g. when the clusters are demographic groups). It would be great if the authors could discuss this.

---

> ### Author Rebuttal · Authors · 2023-08-09
>
> Thank you for taking the time to read and review our paper!
>
> ## General comments on weaknesses:
> 1) We expect that the virtue of look-alike modeling on generalization to apply in a broad range of settings. The high-level intuition is that the look-alike modeling acts as a regularizer and can improve the generalization by reducing the variance, an observation that is not limited to linear regression. To support it empirically, in our **global response** we provide additional experiments for a nonlinear model where we observe similar phenomena.
>
>
> 2) We would like to emphasize that Proposition 3.4 **does not make such an assumption**. Note that the proposition statement involves two matrices $M$ and $\widetilde{M}$. The matrix $M$ corresponds to the matrix with true cluster means as it is columns, and $\widetilde{M}$ is the estimated cluster means. Similarly, $\Lambda$ is the matrix which encodes true cluster memberships, while  $\widetilde{\Lambda}$ is the estimated one. Our proposition allows both estimation $\widetilde{M}$ and $\widetilde{\Lambda}$ be different from the true ones.
>
> 3) Let us make a clarification here. There are two components of the problem setting: (i) data generative model (linear regression and Gaussian Mixture model on the features), and (ii) trained model (Look-alike). The latter is how the learner fits a model to predict $y$ from the features. The response $y$ of course is based on individual features as you said, but the look-alike fits a model by regressing the response $y$ against ($\mu_s$,$x_{{\rm ns}}$) as opposed to ($x_s$, $x_{{\rm ns}}$) (which is done in the vanilla regression). This is what we mean by “look-alike clustering drops the $\langle z_s, \theta_s\rangle$ term”. In other words, it drops this term form the model it uses to regress against the response $y$. At low SNR, this corresponds to removing a `noise’ component from the model ($\langle z_s,\theta_s\rangle$ is of order of the noise $\varepsilon$) and avoids overfitting to this component. We will rewrite this part in the revision to ensure clarity and also include your explanation as well.
>
>
> ## Response to questions:
> 1) The explanation applies to other cases as well (e.g., in case 1, Fig3 we also see higher gain at lower SNR). We initially thought to put the explanation right after the theorem statement, but now we understand the confusion it made and will move it to the beginning of the section.
>
> 2) Line 237: The upperbound on the perturbation is non-negative only if $\psi_d-\psi_p \le 0.5$. Recall that $\psi_d - \psi_p = (d-p)/n$ is the ratio of non-sensitive features on the sample size. For $\psi_d-\psi_p\ge 0.5$ the statement is vacuous  as the assumption is not satisfied. However, note that in this regime (or its extreme where n<< d-p) it is hopeless to have a reasonable estimate of the clusters  as we have substantially smaller sample size than the problem dimension.
> That said, please note that this proposition only makes an assumption on the perturbation norm, and NOT any specific estimator of the clusters (so perturbations can be added in an adversarial way as far as it satisfies the norm constraint).  In other words, the result holds against strong adversarial perturbation. If one instead focuses on specific estimator of the clusters, then the bound on $\delta$ can probably be made more relaxed.
>
> 3) Please see our responses to the weaknesses.
>
> Thanks for pointing out the typos. We will fix them.

---

> > ### Comment · Reviewer_EnfZ · 2023-08-15
> > **Follow-up questions**
> >
> > I thank the authors for their careful response.
> >
> > I think the added experiment is helpful for motivating the problem and demonstrating its impact, and I wonder if the importance of the work can be further highlighted. A strength of the paper is that it gives exact formulas for the expected squared loss, whereas a weakness is that it does not explain well what insight these quantitative formulas give in addition to a qualitative high-level explanation: 1) look-alike clustering reduces the model capacity--> regularization --> generalization (substantially reduced variance) and 2) low SNR implies small loss caused by the reduced model capacity (only slightly increased bias). The qualitative explanation already seems quite convincing for linear regression. For comparison, the double descent phenomenon and the long-tail theory seem very hard to explain convincingly (even in hindsight) without resorting to quantitative analysis.
> >
> > The authors' response on the dropped term $\\langle z_s,\theta_s \\rangle$ makes sense. I think it is important to make the distinction between the training process and the data-generating process (data model (2.1)) clearer in the next version. The term $\\langle z_s,\theta_s \\rangle$ is currently presented in the description of the data-generating process, but it is only meant to be dropped in the training process (thus restricting the capacity of the learned model, which corresponds to regularization).
> >
> > Here are some follow-up questions:
> >
> > **Purtabation assumption in proposition 3.4.** I understand that the assumption on $\\widetilde M_s$ and $\\widetilde \\Lambda$ is only about their product and some amount of error $\\delta_n$ is allowed. My question was whether the authors know of an efficient algorithm that takes the original data set (before look-alike clustering) as input and produces such estimates $\\widetilde M_s$ and $\\widetilde \\Lambda$ that satisfy the assumption.
> >
> > **Over-parameterized version of proposition 3.4.** Based on the authors' response, proposition 3.4 only applies to the significantly under-parameterized regime $\\psi_d - \\psi_p \le 0.5$. Is there a fundamental barrier to extending it to the over-parameterized regime? In the over-parameterized regime, $\\psi_p$ can still be very small, so we do not seem to be restricted by the data set size for the purpose of recovering the cluster means for the sensitive features.
> >
> > **Explanation of improved generalization.** Does the explanation at Line 297 apply to Case 3? Why or why not?

---

> > > ### Author Response · Authors · 2023-08-16
> > >
> > > Thanks for your comments. We will use the added experiment in the revised version to argue that the use of look-alike modeling in generalization goes beyond linear regression. As you said we have qualitative explanation to justify the insights we get from quantitative analysis, in the hindsight. A precise quantitative theory is more helpful in understanding thresholds (like how small should SNR be to see positive gain) and  the key factors on the generalization (eg., the fact that correlation between cluster centers and the model $\theta_0$ is a key factor or how the size/number of clusters effect generalization is only understood through a precise quantitative theory).
> > >
> > > # Response to follow-up questions:
> > > 1) There are many efficient algorithms proposed to learn cluster membership under a Gaussian Mixture model, including semidefinite programs, Lloyd’s Algorithm (a greedy iterative method to approximate K-means), spectral clustering, tensor decompositions, method of moments, among others. In particular (Loffler et.al) shows optimality of spectral method (in terms of sample complexity) for GMM. We have not worked out their derived sample complexity in terms of our specific error measure $||M\Lambda - \tilde{M}\tilde{\Lambda}||$, as it has not been the focus of the paper, but we expect spectral clustering should satisfy the assumption when the cluster centers separation is large enough. (The analysis of (Loffler et.al) allows sample size and features dimension to be of the same order and shows that the fraction of misclustered points goes down exponentially fast in the cluster centers separation. Then we can focus on the correctly clustered ones and use a crude bound for the rest. On the correct ones $\Lambda$ and $\tilde{\Lambda}$ become the same and we just need to bound the distance between $M$ and $\tilde{M}$ which is same as bounding the deviation of sample average of points from the actual average.)
> > >
> > > *Loffler, Zhang, Zhou, Optimality of spectral clustering for Gaussian mixture model, Annals of Statistics, 2021.
> > >
> > > 2) After the rebuttal period, we scrutinized the proof and realized that we can have a similar result for the overparameterized regime, where the assumption in this case reads as $\delta_n\le c(\sqrt{\psi_d-\psi_p-1} - 1)$. So the only part that is not covered is when $0.5< \psi_d-\psi_p<2$. The fact that $\psi_d-\psi_p$ should be a way from 1 is something fundamental, because otherwise the look-alike features will have very small (non-zero) singular value and the adversary can put all the perturbation in that space. Since the ridge-less estimator depends on the pseudo-inverse of the features matrix, this lead to a large perturbation on the estimator. But requiring $\psi_d-\psi_p$ to be outside (0.5,2), we believe is an artifact of the analysis and the result of [28] that we used in our proof.
> > >
> > > 3) Yes, it applies to case 3 as well. Note that explanation (based on regularization act of look-alike modeling) and avoiding overfitting in low-SNR does not assume anything about under/over-paramterized regime. Indeed, we had a figure similar to Fig 3(a) for case 3, which we removed due to space constraint. (Similar to Fig 3(a), it shows a positive gain at low SNR.)

---

> > > > ### Comment · Reviewer_EnfZ · 2023-08-16
> > > >
> > > > Thanks for the response which addresses my questions. I think the results and discussions provided by the authors during the rebuttal phase should lead to improvement of the paper. I'm increasing my score with that prospect.

---

> > > > > ### Author Response · Authors · 2023-08-16
> > > > >
> > > > > Thank you for your invaluable comments and also for raising your score.

---

### Author Rebuttal · Authors · 2023-08-09

We would like to sincerely thank all the reviewers for taking time to review our paper and for the valuable feedbacks. A common comment raised by some of the reviewers was on the limitation of the work and whether the message of our paper goes beyond linear regression.

We expect the virtue of look-alike modeling on generalization to apply in a broad range of settings. The high-level intuition is that the look-alike modeling acts as a regularizer and can improve the generalization by reducing the variance, an observation that is not limited just to linear regression. To support it empirically, here we provide additional experiments for a nonlinear data model. We consider the setting where the response is generated as $y = \exp(X\theta_0 + \varepsilon)$, with $\varepsilon$ Gaussian noise, and the estimators are obtained by fitting a generalized linear model with logarithm link function and poisson distribution. As the plot in the attached pdf shows, at smaller SNR we observe a gain in the generalization of the look-alike estimator over the vanilla estimator which uses individual sensitive features (similar behavior to Fig 3a in the paper).

In terms of theory, a similar approach based on CGMT can be used to handle other loss functions, albeit leading to a much more complicated formula for the generalization, which is less transparent in showing the message. Nonetheless, a limitation of our analysis is the distribution of features, as the CGMT strongly relies on the gaussianity of the features.

---

### Decision · Program_Chairs · 2023-09-21

**Decision:**

Accept (poster)

**Comment:**

The paper studies linear regression where some coordinates of the covariate x are not revealed directly to the learner, and only a cluster-wise average is revealed for those coordinates. This change in the covariates x is called look-alike clustering and it has been used for protecting sensitive attributes in prior work.  The authors show closed-form formulas for the asymptotic performance with and without look-alike clustering. This allows the authors to identify regimes when look-alike clustering improves the performance. They also included experimental results verifying the formulas.